# Structures reveal opening of the store-operated calcium channel Orai

**Xiaowei Hou, Shana R Burstein, Stephen Barstow Long***

Structural Biology Program, Memorial Sloan Kettering Cancer Center, New York, United States

**Abstract** The store-operated calcium ($Ca^{2+}$) channel Orai governs $Ca^{2+}$ influx through the plasma membrane of many non-excitable cells in metazoans. The channel opens in response to the depletion of $Ca^{2+}$ stored in the endoplasmic reticulum (ER). Loss- and gain-of-function mutants of Orai cause disease. Our previous work revealed the structure of Orai with a closed pore. Here, using a gain-of-function mutation that constitutively activates the channel, we present an X-ray structure of *Drosophila melanogaster* Orai in an open conformation. Well-defined electron density maps reveal that the pore is dramatically dilated on its cytosolic side in comparison to the slender closed pore. Cations and anions bind in different regions of the open pore, informing mechanisms for ion permeation and $Ca^{2+}$ selectivity. Opening of the pore requires the release of cytosolic latches. Together with additional X-ray structures of an unlatched-but-closed conformation, we propose a sequence for store-operated activation.
DOI: https://doi.org/10.7554/eLife.36758.001

## Introduction

The dearth of calcium ($Ca^{2+}$) ions in the cytosol of non-excitable metazoan cells under 'resting' conditions allows transient increases in the cytosolic calcium concentration to relay internal messages and enables cells to respond to external stimuli. These cytosolic calcium signals regulate a plethora of processes including gene transcription, cell motility, exocytosis, and cell metabolism (*Clapham, 2007*). Short-lived increases in cytosolic $[Ca^{2+}]$ can be generated by the release of $Ca^{2+}$ into the cytosol from the endoplasmic reticulum (ER) through ion channels such as the IP3R channel (*Clapham, 2007*). A second, more long-lasting elevation in cytosolic calcium occurs by the opening of Orai channels in the plasma membrane that allow $Ca^{2+}$ ions to flow into the cell (*Hogan et al., 2010*). The driving force for $Ca^{2+}$ entry is substantial: the negative voltage inside the cell is an attractive force and in the same direction as the chemical gradient for $Ca^{2+}$ – approximately 2 mM $[Ca^{2+}]$ outside the cell and 20,000–fold lower inside the cell (~100 nM). Despite these driving forces, Orai conducts ions approximately 100-times slower than most $Ca^{2+}$ channels (Orai has a single channel conductance of only ~20 fS [*Prakriya and Lewis, 2006*]), and this probably serves important physiological roles: it would prevent overwhelming the cell with $Ca^{2+}$ when the channel opens and would restrict the $Ca^{2+}$ elevation to a small region around the channel. Since the nature of the open pore is not yet known, it is unclear how Orai limits the rate of $Ca^{2+}$ influx, but the physiochemical environment of the opened pore will undoubtedly be distinct from ion channels with high conductance. Orai is one of the most $Ca^{2+}$-selective channels known and the structure of an opened pore could also provide insight into how the channel achieves such exquisite selectivity.

Calcium influx through Orai is necessary for activation of immune response genes in T cells and a range of other physiological processes (*Feske et al., 2005*; *Lacruz and Feske, 2015*; *Prakriya and Lewis, 2015*). Mutations in Orai have been implicated in a spectrum of maladies. Generally, loss-of-function mutations cause immune system dysfunction, including severe combined immunodeficiency-like disorders (*Feske et al., 2006*; *Lacruz and Feske, 2015*). Gain-of-function mutations of

*For correspondence:
longs@mskcc.org

**Competing interests:** The authors declare that no competing interests exist.

Orai have also been identified. These result in constitutive activation of the channel and have been associated with tubular aggregate myopathy and Stormorken syndromes (*Lacruz and Feske, 2015*).

There are three Orai proteins in humans (Orai1-3). *Drosophila melanogaster* contains one ortholog (Orai), which shares 73% sequence identity to human Orai1 in the transmembrane region, and is the most studied non-human Orai channel. The channels have broad tissue distribution and are tightly regulated (*Hogan et al., 2010*). In the quiescent state before activation, the ion pore of Orai is closed to prevent aberrant $Ca^{2+}$ flux through the plasma membrane. The channel is activated by the depletion of $Ca^{2+}$ from the endoplasmic reticulum (ER), and as such it was characterized as the calcium release-activated calcium (CRAC) channel responsible for store-operated calcium entry (SOCE) before the molecular components were known (*Hoth and Penner, 1992*). Orai was identified as the protein that forms the channel's pore and STIM was identified as its regulator (*Feske et al., 2006*; *Liou et al., 2005*; *Prakriya et al., 2006*; *Roos et al., 2005*; *Vig et al., 2006a*; *Yeromin et al., 2006*; *Zhang et al., 2006*; *Zhang et al., 2005*). Recent studies have uncovered the general mechanism of channel activation, which is distinct from the activation mechanisms known for other channels (reviewed in *Hogan and Rao, 2015*; *Prakriya and Lewis, 2015*). As a result of depletion of $Ca^{2+}$ within the ER, STIM, which is a single-pass membrane protein resident to the ER, localizes to regions where the ER and plasma membranes are separated by only 10–20 nM. Here, STIM physically interacts with cytosolic regions of Orai to open its pore. We previously determined the X-ray structure of *Drosophila melanogaster* Orai in a 'quiescent' conformation with a closed ion pore (*Hou et al., 2012*). The conformational changes that lead to opening and the conformation of the opened pore are unknown.

The X-ray structure of the quiescent conformation provides a foundation to understand the molecular basis for the function of Orai (*Hou et al., 2012*). The channel is formed from an assembly of six Orai subunits that surround a single ion pore, which is perpendicular to the plasma membrane in a cellular setting (*Figure 1A*) (*Hou et al., 2012*). Although the oligomeric state revealed by the X-ray structure was a surprise, further studies have shown that the functional state of human Orai1 is also as a hexamer of subunits (*Cai et al., 2016*; *Yen et al., 2016*). Each Orai subunit contains four transmembrane helices (M1-M4) and a cytosolic M4-ext helix (*Figure 1*). Amino acid side chains on the M1 helices from the six

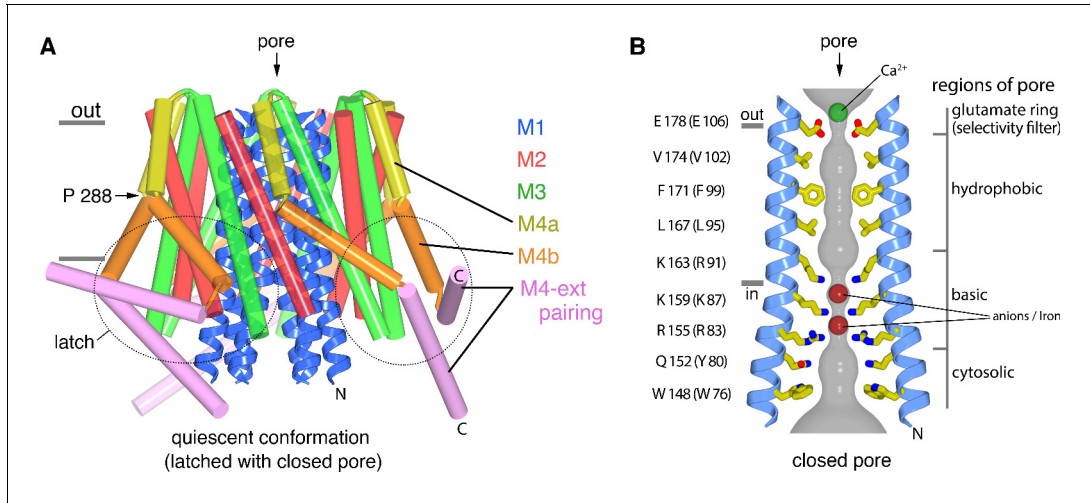

**Figure 1.** The quiescent conformation –latched with a closed pore. (a) Overall structure of Orai, from PDB ID 4HKR, in a 'quiescent' conformation (*Hou et al., 2012*). The pore is closed; M4-ext helices pair with one another. The M1 helices are depicted as blue ribbons; other helices are cylinders. The approximate boundaries of the plasma membrane are shown as gray bars. Regions referred to as 'latches' in this study are indicated as dashed ovals. (b) Close-up view of the closed pore. Two M1 helices are drawn as ribbons (four M1 helices are omitted for clarity). The pore is a depicted as a gray surface indicating the minimal radial distance to the nearest van der Waals contact. Amino acid side chains that form the walls of the pore are drawn as sticks and colored (yellow for carbon, blue for nitrogen, and red for oxygen). Amino acid numbering is shown for *Drosophila melanogaster* Orai without parentheses and for human Orai in parentheses. Sections of the pore are indicated. Horizontal gray bars correspond to the approximate boundaries of the membrane, although the M1 helices are shielded from the membrane by M2 and M3. A $Ca^{2+}$ ion is indicated. Red spheres mark the location of anomalous difference electron density attributed to iron, perhaps bound as $(FeCl_6)^{3-}$ (*Hou et al., 2012*). The complex anion $(IrCl_6)^{3-}$ also binds in these sites (*Hou et al., 2012*).

DOI: https://doi.org/10.7554/eLife.36758.002

subunits form the walls of the pore (*Figure 1B*). In contrast to many ion channels, amino acid side chains establish the dimensions and chemical environment along the entirety of the pore. The M2 and M3 helices form a shell surrounding the M1 helices and shield them from the membrane. The M4 helices are located at the periphery and contain two segments, M4a and M4b, delineated by a bend at a conserved proline residue (Pro288). Following M4b, the M4-ext helices extend into cytosol. The M4-ext helices from neighboring subunits interact with one another through pairwise helical coiled-coil packing, which creates a belt-like arrangement surrounding the channel on its intracellular side (*Figure 1A*). Mutation of the hydrophobic residues that mediate the coiled-coils has been shown to prevent channel activation by STIM, possibly by reducing the affinity for STIM (*Muik et al., 2008*; *Navarro-Borelly et al., 2008*). Because the cytosolic domain of STIM also contains coiled-coil regions (*Fahrner et al., 2014*; *Stathopulos et al., 2013*; *Yang et al., 2012*), it has been proposed that STIM interacts with the M4-ext helices through a coiled-coil interaction and that the structure represents a quiescent conformation prior to the binding of STIM (*Hou et al., 2012*).

How STIM binding causes the channel to open is an intense area of research, and many questions remain including those as fundamental as how many STIM molecules bind to the Orai channel in the activated complex (reviewed in *Prakriya and Lewis, 2015*). An NMR study of a complex between a portion of the M4-ext helix of Orai and a small cytosolic fragment of STIM containing its coiled-coil regions 1 and 2 (CC1 and CC2) provides the only three-dimensional structural information for the complex of these proteins to date (*Stathopulos et al., 2013*). However, the relevance of this structure to the full assembly is unclear, in part because the portion of STIM in the complex is missing a key region (coiled coil region 3; CC3) that is necessary for channel activation (*Kawasaki et al., 2009*; *Park et al., 2009*; *Yuan et al., 2009*) and in part because data suggest that the N-terminal region of M1 also interacts with STIM (*Lis et al., 2010*; *Park et al., 2009*). Ultimately, because both the structure of Orai and its mode of activation are so distinct from the structures and activation mechanisms that are understood for other channels, structural information on the Orai-STIM complex using full-length proteins will be most informative for understanding how STIM activates the channel on a molecular level.

The X-ray structure of the quiescent conformation of Orai revealed that the closed pore is approximately 55 Å long, narrow, and impermeant to ions (*Figure 1B*) (*Hou et al., 2012*). It contains four sections: a glutamate ring on the extracellular side that forms the selectivity filter (comprising Glu178 residues from the six subunits), a ~15 Å-long hydrophobic section, a ~15 Å-long basic section, and a cytosolic section (*Figure 1B*). Mutation of the residue corresponding to Glu178 in human Orai1 (Glu106) to aspartate disrupts $Ca^{2+}$-selectivity (*Prakriya et al., 2006*; *Vig et al., 2006b*; *Yeromin et al., 2006*). The walls of the basic section are formed by three amino acids from each of the six M1 helices that are conserved as lysine or arginine in Orai channels (*Figure 1B*). Mutation of the top basic residue in human Orai1 (R91W, corresponding to K163W in *Drosophila* Orai) causes a severe combined immune deficiency-like disorder by preventing channel activation (*Derler et al., 2009*; *Feske et al., 2006*). The presence of 18 basic residues (three from each of the six subunits) within the pore of a cation channel is highly unusual and would presumably establish an electrostatic barrier that opposes $Ca^{2+}$ permeation in the closed state. We found that the closed conformation of the basic region serves as a binding site for anion(s) that stabilize the arginine/lysine residues in close proximity (*Hou et al., 2012*). In the structure, an iron complex that co-purifies with the channel, which may be $(FeCl_6)^{3-}$, binds at the center of the basic residues like a plug. While the physiological ligand may or may not be an iron complex, it would seem that the plug would need to be displaced to allow $Ca^{2+}$ permeation through the pore when the channel is open.

A structure of Orai with an open pore would markedly advance our understandings of the channel but structural studies of the complex between Orai and STIM are complicated by the low affinity of the interaction and because the stoichiometry between Orai and STIM is not fully established. Gain-of-function mutations of Orai provide a potential experimental avenue for capturing a structure of an open pore because STIM would not necessarily be required for channel activation. Certain mutations of M1 residues that line the pore create constitutively active channels but these channels have reduced selectivity for $Ca^{2+}$ in comparison to STIM-activated Orai, which suggests that their pores have non-native conformations (*McNally et al., 2012*; *Yamashita et al., 2017*; *Zhang et al., 2011*). The H134A mutation of human Orai1, on the other hand, does not line the pore and has been shown to generate an activated channel with high selectivity for $Ca^{2+}$ in the absence of STIM (*Frischauf et al., 2017*; *Yeung et al., 2018*). The high selectivity and the normal inwardly rectifying current-voltage relationship of the H134A mutant suggest that this mutation induces a conformation

of the pore that is highly similar to the conformation of the pore when Orai is activated by STIM (*Frischauf et al., 2017*; *Yeung et al., 2018*). In this study, we introduced the corresponding amino acid substitution into *Drosophila* Orai (H206A), confirmed that it generates a constitutively active channel, and determined its X-ray structure. The structure reveals a dilated pore and conformational changes in cytosolic regions that must 'unlatch' for the pore to open. In another set of experiments, we determined structures of the wild-type channel and of a mutant that corresponds to one that causes immune deficiency in humans. These structures reveal an 'unlatched-closed' conformation that is a structural hybrid of the quiescent and open conformations, and this indicates that unlatching, while necessary, does not necessarily cause pore opening. In additional experiments, we investigated ion-binding properties of the open pore to gain insight into $Ca^{2+}$-selectivity and block by trivalent lanthanides. The studies reveal mechanisms for pore opening and give insight into the basis of store-operated $Ca^{2+}$ entry.

## Results

### Constructs of Orai used for structural studies function as CRAC channels

Orai from *Drosophila melanogaster* (hereafter referred to as Orai) was selected for functional and structural studies on the basis of its good biochemical stability in detergent-containing solutions

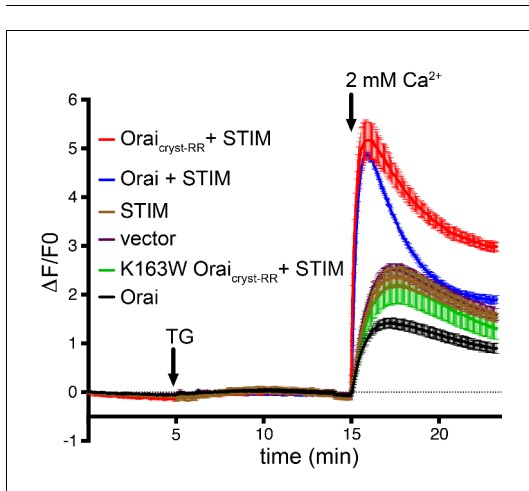

**Figure 2.** $Ca^{2+}$ influx measurements show that *Drosophila* Orai constructs function as CRAC channels when co-expressed with STIM. HEK293 cells were transfected with Orai and/or STIM as indicated (also see Materials and methods). Cytosolic $[Ca^{2+}]$ levels were detected using the genetically-encoded fluorescent $Ca^{2+}$ indicator GCaMP6s (*Chen et al., 2013*); data are plotted as the change in fluorescence intensity relative to the initial value (ΔF/F0) versus time. Thapsigargin (TG), which is used to deplete ER calcium stores, and 2 mM $CaCl_2$ were added at the indicated times (arrows). $Ca^{2+}$ influx above the background level (vector alone) was observed for the co-expression of STIM with $Orai_{cryst-RR}$ or wild-type Orai, but not for the constitutively closed K163W mutant or for Orai or STIM alone. Signal from endogenous CRAC channels is apparent from the vector control (empty expression vector). Standard error, derived from three independent measurements, is shown.
DOI: https://doi.org/10.7554/eLife.36758.003

(*Hou et al., 2012*). The previously determined X-ray structure of Orai in the quiescent conformation was obtained using construct that contained the regions necessary for STIM-activation but its ability to be activated by STIM had not been evaluated (*Hou et al., 2012*; *Li et al., 2007*). In that construct (herein denoted '$Orai_{cryst-RR}$'), a poorly conserved N-terminal region and a few C-terminal amino acids were removed, and two proline residues were replaced by arginine in the hypervariable M3-M4 loop (P276R and P277R mutations) in order to obtain well-diffracting crystals. The structures presented here are based on a nearly identical construct (herein denoted '$Orai_{cryst}$') that only differs from $Orai_{cryst-RR}$ in that the two M3-M4 residues are present as their wild-type proline counterparts. To assess the potential impact of the P276R/P277R mutations on the function of the channel, we co-expressed Orai and STIM in HEK239 cells and studied $Ca^{2+}$ influx (*Figure 2*). We find that both the previously crystallized construct and one with a wild-type M3-M4 loop operate as CRAC channels that can be activated by STIM (*Figure 2*). Thus, the constructs used for crystallization possess the fundamental properties of previously studied Orai channels.

### Activity of H206A Orai

We introduced the H206A mutation, which corresponds to gain-of-function H134A mutation of human Orai1 (*Frischauf et al., 2017*; *Yeung et al., 2018*), into Orai and studied the purified channel (H206A $Orai_{cryst}$) in proteoliposomes to assess activity (*Figure 3*). Using divalent-free conditions, under which ionic currents through CRAC channels are more easily observed

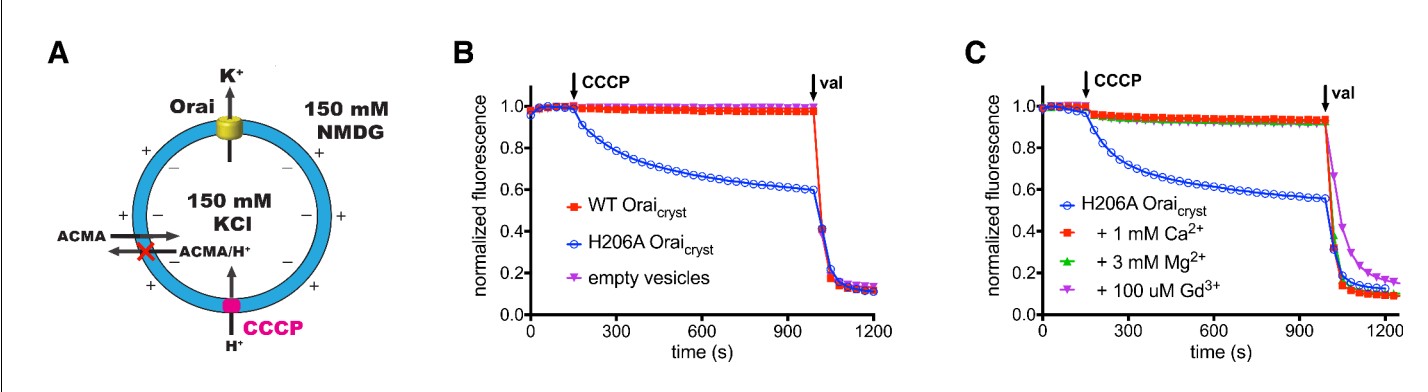

**Figure 3.** Ion flux through H206A Orai_cryst in liposomes. (a) Schematic of the fluorescence-based flux assay. Vesicles containing WT or H206A Orai_cryst or those prepared without protein (empty vesicles) were loaded with 150 mM KCl and were diluted 50-fold into flux buffer containing a fluorescent pH indicator (ACMA) and 150 mM N-methyl-D-glucamine (NMDG) to establish a $K^+$ gradient (Materials and methods). After stabilization of the fluorescence signal (150 s), a proton ionophore (CCCP) was added. An electric potential arising from $K^+$ efflux was used to drive the uptake of protons, which quenches the fluorescence of ACMA. A red 'X' indicates that ACMA is not membrane-permeable in the protonated form. (b) $K^+$ flux measurements for WT and H206A Orai_cryst. The time-dependent decrease in fluorescence observed for H206A Orai_cryst after the addition of CCCP is indicative of $K^+$ flux. Valinomycin (val) was added after 990 s to render all vesicles permeable to $K^+$ and establish a baseline fluorescence. Traces were normalized by dividing by the initial fluorescence value, which was within ±10% for each experiment. (c) $K^+$ flux through H206A Orai_cryst is inhibited by $Ca^{2+}$, $Mg^{2+}$ and $Gd^{3+}$.

DOI: https://doi.org/10.7554/eLife.36758.004

due to greater conductance of monovalent cations (e.g. $Na^+$ or $K^+$) than $Ca^{2+}$ (*Lepple-Wienhues and Cahalan, 1996*; *Prakriya and Lewis, 2006*), we observed robust $K^+$ flux through the channel. Ion flux was not observed for empty vesicles or through a channel without the H206A mutation (WT Orai_cryst), as is expected without activation by STIM (*Figure 3B*). Similar to wild-type CRAC channels, $K^+$ flux was blocked by the addition of $Gd^{3+}$ (*Figure 3C*) (*Yeromin et al., 2006*). $K^+$ flux through H206A Orai_cryst was also inhibited by the addition of $Mg^{2+}$ or $Ca^{2+}$ (*Figure 3C*), which is in accord with the properties exhibited by STIM-activated channels and indicative of the channel's selectivity for $Ca^{2+}$ (*Lepple-Wienhues and Cahalan, 1996*; *Prakriya and Lewis, 2006*). Thus, as has been shown for the corresponding H134A mutation of human Orai1 (*Frischauf et al., 2017*; *Yeung et al., 2018*), H206A Orai_cryst forms an open channel that recapitulates properties of STIM-activated Orai.

## X-ray structure of H206A Orai_cryst reveals an open conformation

Obtaining X-ray structural information for Orai has been challenging and visualizing an open conformation of the pore especially so. Extensive optimization of crystallization conditions improved the quality of H206A Orai_cryst crystals from an initial diffraction limit of 20 Å resolution to 6.7 Å resolution. Despite the modest resolution of the optimized crystals, we were able to discern the conformation of the channel by calculating electron density maps using non-crystallographic symmetry averaging, which can be applied when there are multiple copies of the polypeptide in the crystallographic asymmetric unit (*Bricogne, 1974*). In this case, the asymmetric unit contains 24 Orai subunits, which are arranged as four complete channels. The 24-fold non-crystallographic symmetry allowed us to accurately determine the crystallographic phases and obtain electron density maps of excellent quality, which delineate all α-helices of the channel and resemble maps calculated using considerably higher resolution diffraction data (*Figure 4A,B*, *Figure 4—video 1*, Materials and methods). All four channels in the asymmetric unit adopt the same conformation. Since side chains are not visible in the maps, we collected a highly redundant dataset using an X-ray wavelength (λ = 1.7085 Å) that was chosen to optimize the anomalous diffraction signal from endogenous sulfur atoms in order to locate methionine and cysteine residues within the protein (*Table 1*). Anomalous-difference electron-density peaks corresponding to these amino acids indicate both the validity of the atomic model and the accuracy of the crystallographic phases that were used to generate the electron density maps of the channel (*Figure 4—figure supplement 1*).

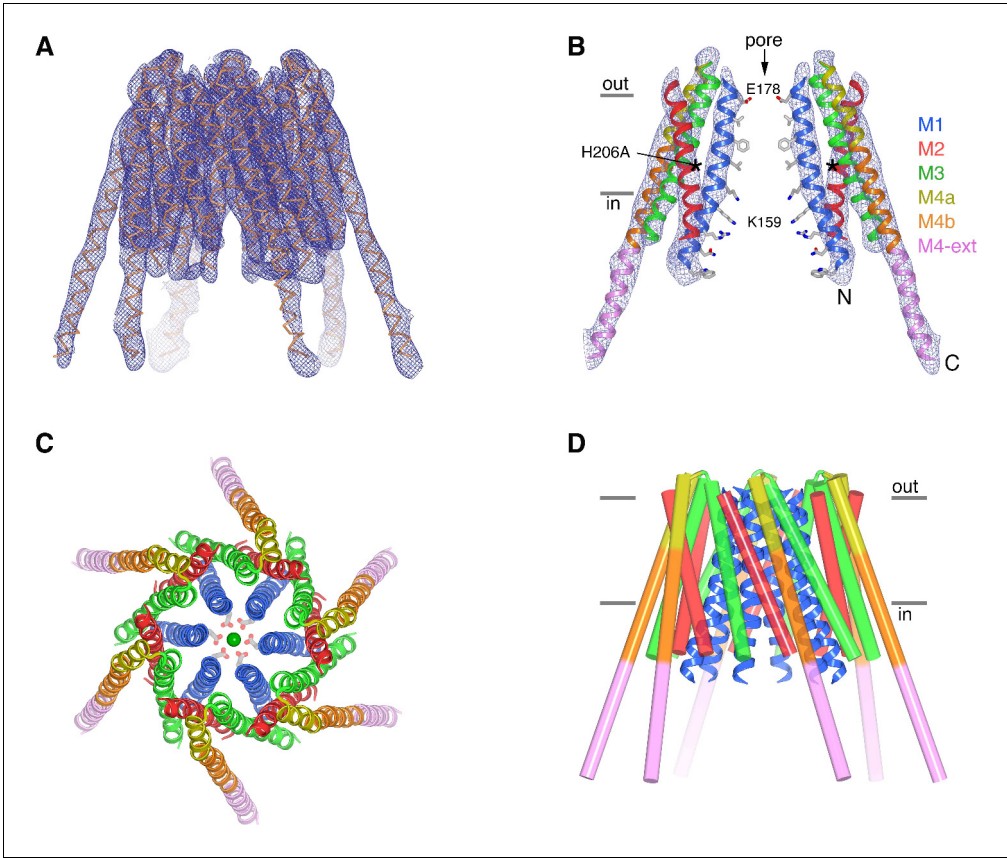

**Figure 4.** The structure of H206A Orai$_{cryst}$ reveals an open conformation.  (a) Electron density map of H206A Orai$_{cryst}$. The map (blue mesh, contoured at 1.4 σ, and covering one channel) was calculated from 20 to 6.7 Å using native-sharpened amplitudes and phases that were improved by 24-fold non-crystallographic symmetry (NCS) averaging, solvent flattening and histogram matching (Materials and methods). The atomic model is shown in Cα representation. *Figure 4—video 1* shows a video of this Figure. (b) Side view showing two opposing subunits of H206A Orai$_{cryst}$ and the same electron density map. Asterisks mark the location of the H206A substitution. Amino acid side chains on the pore are shown only for reference (sticks). Approximate boundaries of the membrane are shown as horizontal bars. Helices are depicted as ribbons and colored as indicated. (c) Extracellular view showing the hexameric architecture. Helices are depicted as ribbons, with Glu178 side chains (sticks) and Ca$^{2+}$ ion (green sphere) shown for reference. (d) Overall structure, shown in the same orientation as (a). The M1 helices are drawn as blue ribbons and the other helices are shown as cylinders.

DOI: https://doi.org/10.7554/eLife.36758.005

The following video and figure supplement are available for figure 4:

**Figure supplement 1.** Anomalous-difference electron-density at cysteine and methionine residues in the final model of H206A Orai$_{cryst}$, depicted in stereo.

DOI: https://doi.org/10.7554/eLife.36758.006

**Figure 4—video 1.** Electron density for H206A Orai$_{cryst}$ from *Figure 4A*.

DOI: https://doi.org/10.7554/eLife.36758.007

The X-ray structure of H206A Orai$_{cryst}$ reveals a new conformation of the channel with a dilated pore that presumably represents an open conformation (*Figure 4*). The open channel is composed of a hexameric assembly of Orai subunits surrounding a single ion pore (*Figure 4*). The overall architecture of the channel is similar to the quiescent conformation, with each Orai subunit containing four transmembrane helices (M1-M4). The six M1 helices, one contributed by each subunit of the channel, form the walls of the open pore. Because the secondary structure of the polypeptide surrounding the pore is α-helical, amino acid side chains on M1 establish the chemical environment along the pore. In comparison to the closed pore (*Hou et al., 2012*), the pore is dramatically dilated on its cytosolic end, expanding by ~10 Å at Lys159 (*Figure 5*). The differences between the closed

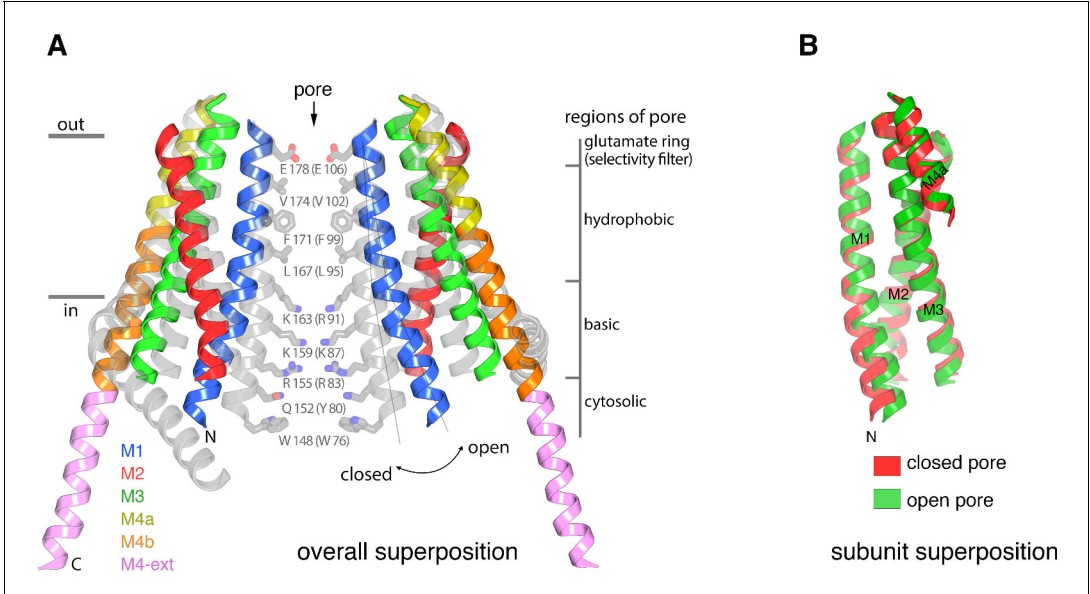

**Figure 5.** Conformational changes between the quiescent and open conformations. (a) Superimposed structures of the quiescent (PDB ID: 4HKR) and open (H206A Orai$_{cryst}$) conformations are drawn in ribbon representation. Two opposing subunits are shown, surrounding the pore, with the open conformation colored as indicated and the closed conformation in gray. Thin lines and a curved arrow highlight the outward rotation of subunits (with its fulcrum near Glu178) and the slight additional bend in M1. Conformational changes of M4/M4-ext are also apparent. Amino acids forming the walls of the closed pore (from the quiescent conformation) are shown as sticks, with corresponding regions of the pore indicated. Amino acids in parentheses denote human Orai1 counterparts. Horizontal bars indicate approximate boundaries of the plasma membrane. (b) Comparison of M1-M4a from individual subunits between the quiescent and open conformations. The region of an Orai subunit spanning M1 through M4a was superimposed between the quiescent (red ribbons, PDB ID 4HKR) and open (green ribbons, H206A Orai$_{cryst}$) conformations. The slight additional bend in M1 of the open conformation is apparent at its N-terminal end. Otherwise the M1-to-M4b region of the two subunits superimpose within the error of the coordinates of the open conformation (the root-mean-squared deviation for the C$\alpha$ positions of residues 163 to 288 superimposed in this manner is 1.1 Å).

DOI: https://doi.org/10.7554/eLife.36758.009

The following figure supplement is available for figure 5:

**Figure supplement 1.** Residue 206 in closed and open pores.

DOI: https://doi.org/10.7554/eLife.36758.010

and open pores taper off toward the extracellular side such that, while subtle changes may occur when the pore opens, the location of the M1 helix at Glu178 is indistinguishable from the closed conformation at this resolution.

The electron density indicates that the principal conformational change in the pore results from an outward rigid body rotation of the M1-M4a portion of each subunit away from the central axis and a slight additional outward bend of M1 on its intracellular half (*Figure 5A*). The packing of M1-M4a within an individual subunit is nearly indistinguishable from the packing in the closed conformation (*Figure 5B*). The rigid body motion suggests that the amino acids on M1 that form the walls of the pore in the closed conformation also do so in the open conformation *Figure 4B* and *Figure 5A*). The exposure of these residues to the ion pore in its open conformation is consistent with accessibility experiments of cysteine residues introduced into the M1 helix by mutagenesis (*McNally et al., 2009*; *Zhou et al., 2010*), and this is an indication that the pore of the H206A Orai$_{cryst}$ structure represents the conformation of the pore when the channel is activated by STIM, although further functional and structural analyses are warranted. Because we cannot visualize amino acid side chains due to the limited resolution of the structure, the data are silent on whether opening also involves a slight (~20°) rotation along the helical axis of M1 as has been suggested by electrophysiological studies using cysteine mutations (*Yamashita et al., 2017*). On the basis of the rigid body motion from the high-resolution structure of the quiescent conformation with a closed pore, the walls of the open pore would have four sections: a glutamate ring on the extracellular side that forms the selectivity

filter (comprising Glu178 residues from the six subunits), a ~ 15 Å-long hydrophobic section, a ~ 15 Å-long basic section, and cytosolic section (*Figure 5A*).

Residue 206 is located on the M2 helix and does not line the pore (*Figure 4B*). In the quiescent conformation, the wild-type histidine at this position forms a hydrogen bond with the side chain of Ser165, which is located on the side of M1 facing away from the pore (*Figure 5—figure supplement 1A*) (*Hou et al., 2012*). On the basis of the current structure, a histidine could be accommodated in the open conformation without steric interference, suggesting that the conformation of the pore observed for H206A Orai$_{cryst}$ could be adopted by wild-type Orai (e.g. when activated by STIM, *Figure 5—figure supplement 1B*). In accord with previous studies (*Frischauf et al., 2017*; *Yeung et al., 2018*), we surmise that interactions between non-pore-lining regions of the channel (e.g. M2-M1 interactions) influence pore opening.

## Cation binding in the open pore

To investigate potential binding sites for cations in the open pore that underlie $Ca^{2+}$-selectivity and block of the channel by trivalent lanthanides, we collected X-ray diffraction data from crystals of H206A Orai$_{cryst}$ containing $Gd^{3+}$, which blocks the channel from the extracellular side (*Aussel et al., 1996*; *Yeromin et al., 2006*), and from crystals containing $Ba^{2+}$, which is a permeant surrogate for $Ca^{2+}$ (*Hoth, 1995*) that is more easily identified crystallographically. Anomalous-difference electron density maps, which pinpoint the location of these ions, contained strong density for $Gd^{3+}$ and for $Ba^{2+}$ in the selectivity filter (*Figure 6A,B*). Due to the limited resolution of the diffraction data, the electron densities could represent one or two $Ba^{2+}/Gd^{3+}$ ions that directly coordinate the side chains of the glutamate ring (Glu178 residues from the six subunits). The presence of $Ba^{2+}$ and $Gd^{3+}$ at this location provide evidence that the Glu178 side chains are oriented toward the pore when it is open. $Ca^{2+}$-binding in this region likely underlies Orai's high selectivity for $Ca^{2+}$. Block of the open pore by $Gd^{3+}$ appears to occur by competitive binding within the selectivity filter.

We showed previously that $Gd^{3+}$, $Ba^{2+}$ and $Ca^{2+}$ bind near the glutamate ring when the pore is closed (*Hou et al., 2012*). While the positioning of $Gd^{3+}$ is very similar between the open and closed pores, the positioning of $Ba^{2+}/Ca^{2+}$ is noticeably different (*Figure 6*, *Figure 6—figure supplement 1*). In the closed pore, the $Ba^{2+}/Ca^{2+}$ ion binds on the extracellular side of the selectivity filter, approximately 4 Å above the ring of glutamates, whereas in the open pore, the electron density is located within the glutamate ring rather than above it (*Figure 6*, *Figure 6—figure supplement 1*). The limits of the diffraction data prevent us from discerning differences in the conformations of the glutamate side chains between the open and closed pores but the apparent repositioning of $Ba^{2+}$ is

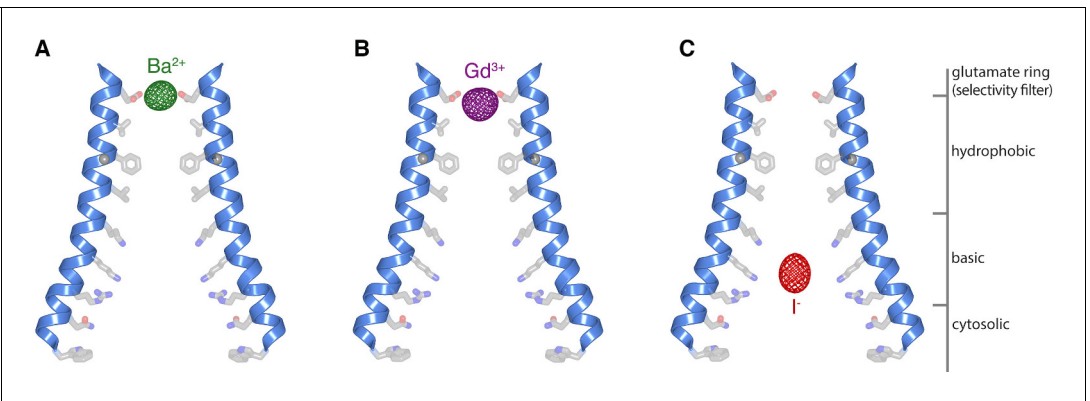

**Figure 6.** Ion binding in the open pore. (a–c) Anomalous-difference electron density maps (mesh) for crystals of H206A Orai$_{cryst}$ with $Ba^{2+}$ (a), $Gd^{3+}$ (b), and I⁻ (c). M1 helices of two opposing subunits are shown as ribbons. Side chains proposed to line the pore (sticks) are drawn for reference; their conformations are hypothetical. The maps are contoured at 10 σ and calculated from 25 to 9 Å for (a–b), and at 7 σ and calculated from 25 to 10 Å for (c).
DOI: https://doi.org/10.7554/eLife.36758.011

The following figure supplement is available for figure 6:

**Figure supplement 1.** Comparison of the anomalous-difference electron-density peaks for $Ba^{2+}$ and $Gd^{3+}$ in the open and closed pores.
DOI: https://doi.org/10.7554/eLife.36758.012

an indication that subtle changes within the selectivity filter occur when the pore opens. Subtle changes at the extracellular side of the pore have also been suggested by spectroscopic and electro-physiological studies when Orai is activated by STIM (*Gudlur et al., 2014*; *McNally et al., 2012*). We conclude that the transition in the pore between non-conductive and conductive conformations involves conformational changes along the length of the pore that introduce functionally important free-energy differences that affect $Ca^{2+}$ selectivity and ion flow. These changes are most structurally pronounced at the cytoplasmic side but they extend energetically to the selectivity filter on the extracellular side.

## Anion binding in the open pore

The basic region of the pore is highly unusual for a cation channel. We have shown previously that the basic region of the closed pore binds anions and that the crystallized protein contains an iron complex within the basic region that co-purifies with the channel (*Hou et al., 2012*). Anomalous difference electron density for iron is not observed in the structure of H206A Orai$_{cryst}$, indicating that the iron complex is not present in the open pore, which is in accord with the dramatic widening of the basic region. To investigate the possibility that anions could bind in the basic region in the open pore, we collected diffraction data from H206A Orai$_{cryst}$ that was crystallized in iodide (I⁻). I⁻ has similar properties to the cellularly abundant Cl⁻ anion and would be identifiable by its anomalous X-ray scattering. We observed robust anomalous difference electron density for I⁻ that is centrally located within the basic region of the open pore (*Figure 6C*). The presence of I⁻ there provides evidence that the basic amino acids are exposed to the pore and that anion(s) can bind in the basic region when it is open. We suspect that a few anions would coat the sides of the basic region in a cellular context. In the open conformation, the basic region is large enough to accommodate a centrally located $Ca^{2+}$ ion that is surrounded by anions and/or water molecules. (The Cα positions of Lys$^{159}$ residues on opposite sides of the pore are ~24 Å apart). We hypothesize that cellular anions may shield the positive charge of the basic residues during the permeation of $Ca^{2+}$ through the open pore.

## Mutation of the basic region

Because the open structure reveals substantial dilation of the basic region, we wondered what effect mutation of the basic residues would have and whether their substitution with serine would create a constitutively open channel. We simultaneously substituted all three basic residues with serine (R155S, K159S, K163S) and studied the purified channel (designated SSS Orai$_{cryst}$) in proteolipo-somes using an assay to measure Na⁺ flux under divalent-free conditions (*Figure 7A*). We chose serine because it is a small hydrophilic residue that would eliminate positive charge from this region and because the R91S mutation in human Orai1 (corresponding to K163S in Orai) forms a functional channel when expressed with STIM1 (*Derler et al., 2009*). However, we did not detect ion flux through SSS Orai$_{cryst}$, suggesting that is not constitutively open (*Figure 7A*). This does not appear to be due to gross misfolding of the channel because protein solubilized by detergent from the proteo-liposomes exhibits a monodisperse size-exclusion chromatography (SEC) profile that is analogous to the profile of protein that was used to determine the X-ray structures (*Figure 7—figure supplement 1A*). Furthermore, we find that Orai channels with the R155S/K159S/K163S (SSS) mutations do not form functional CRAC channels when co-expressed with STIM in HEK293 cells (*Figure 7B*). Flow cytometry using an antibody that binds to the extracellular side of Orai demonstrates that SSS Orai is expressed in the plasma membrane of the cells at a similar level to wild type Orai and SEC suggests that SSS Orai is properly folded in the cells (*Figure 7—figure supplement 1B,C,D*). The inability of SSS Orai to be activated by STIM could be due to constitutive closure of the pore and/or disruption of the interaction with STIM. These somewhat surprising results are an indication of the importance of the basic region in the function of the channel and provide fodder for further study.

## Conformations of M4 and M4-ext identify Pro288 and SHK hinges

Other differences between the quiescent and open conformations are changes in the conformations of the M4 and M4-ext helices. In the quiescent conformation, M4 and M4-ext form three helical seg-ments: M4a and M4b, delineated by a bend in M4 at Pro288 near the midpoint of the membrane, and M4-ext, which follows a bend in a Ser306-His307-Lys308 ('SHK') motif between M4b and M4-ext

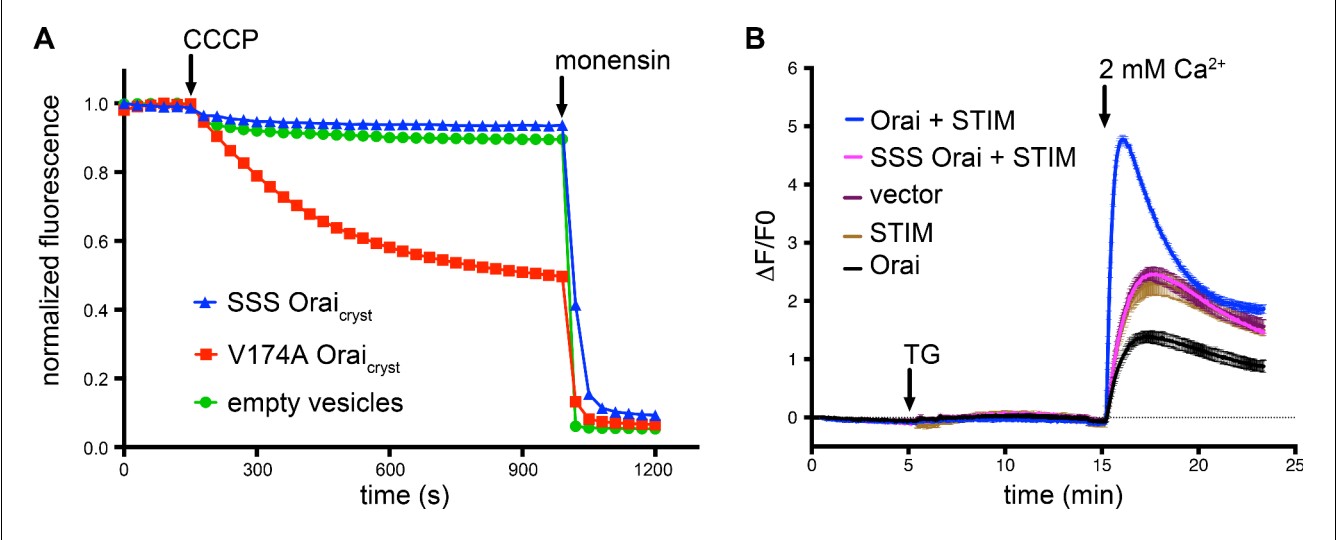

**Figure 7.** SSS Orai$_{cryst}$ is neither constitutively active nor forms functional CRAC channels when expressed with STIM. (a) Proteoliposome-based sodium (Na$^+$) flux assay. Purified proteins were reconstituted into liposomes to assay for Na$^+$ flux under divalent-free conditions (Materials and methods) as described previously (**Hou et al., 2012**). After stabilization of the fluorescence signal (150 s), the proton ionophore CCCP was added to the sample. A decrease in fluorescence is indicative of Na$^+$ flux out of the proteoliposomes. The Na$^+$ ionophore monensin was added after 990 s to render all vesicles permeable to Na$^+$ and establish the minimum baseline fluorescence. The traces were normalized to the initial fluorescence value, which was within ±10% in the experiments. The signal for SSS Orai$_{cryst}$ (Orai$_{cryst}$ with the R155S, K159S, and K163S mutations) is comparable to what is observed for liposomes without protein ('empty vesicles'). As a control for the assay, we observed Na$^+$ flux through purified channels containing the V174A mutation of the hydrophobic region of the pore, which has previously been shown to produce leaky channels with diminished selectivity for Ca$^{2+}$ (**Hou et al., 2012**; **McNally et al., 2012**). (b) Ca$^{2+}$ influx measurements in HEK293 cells expressing indicated Orai and STIM constructs. Methods are identical to those for **Figure 2**. SSS Orai does not show Ca$^{2+}$ uptake above the background level (vector control).

DOI: https://doi.org/10.7554/eLife.36758.013

The following figure supplement is available for figure 7:

**Figure supplement 1.** SSS Orai is properly folded and expressed on the plasma membrane.

DOI: https://doi.org/10.7554/eLife.36758.014

(**Figure 1A**). In the quiescent conformation, the M4-ext helices pair with one another through an antiparallel coiled-coil interaction (**Figure 1A**). In the H206A Orai$_{cryst}$ structure, M4b and M4-ext are repositioned by straightenings of both bends such that the regions corresponding to M4a, M4b and M4-ext of each subunit form a continuous α-helix that traverses the membrane and extends ~45 Å into the cytosolic space (**Figure 4B,D** and **Figure 5A**). The conformational changes of the M4 and M4-ext helices identify the Pro288 residue and the SHK motif, both of which are conserved in Orai channels, as hinges.

## Additional X-ray structures reveal an unlatched-closed conformation

In the crystal of H206A Orai$_{cryst}$, the cytosolic sides of two channels face one another and the M4-ext helices of different channels interact through anti-parallel coiled-coils (**Figure 8A**). These coiled-coil interactions are analogous to the pairing of M4-ext helices between adjacent subunits in the quiescent conformation (**Figure 8—figure supplement 1**). To exclude the possibility that the crystal contacts in the H206A Orai$_{cryst}$ structure were responsible for the conformational changes we observed in the pore, we determined the structures of wild-type (WT) Orai$_{cryst}$ and K163W Orai$_{cryst}$ grown in the same crystal form (I4$_1$). The K163W mutation corresponds to R91W in human Orai1, which is a loss of function mutation that causes a severe combined immune deficiency-like disorder (**Feske et al., 2006**). Well-defined electron density maps of WT and K163W Orai$_{cryst}$ were obtained using non-crystallographic symmetry averaging of modest (6.9 and 6.1 Å, respectively) resolution diffraction data in the same manner as for the H206A Orai$_{cryst}$ structure (**Figure 9A,B**). Anomalous-difference electron density peaks for sulfur atoms of methionine and cysteine residues in WT Orai$_{cryst}$ confirm the accuracy of the atomic model (**Figure 9E**). We also obtained crystals of K163W Orai$_{cryst}$

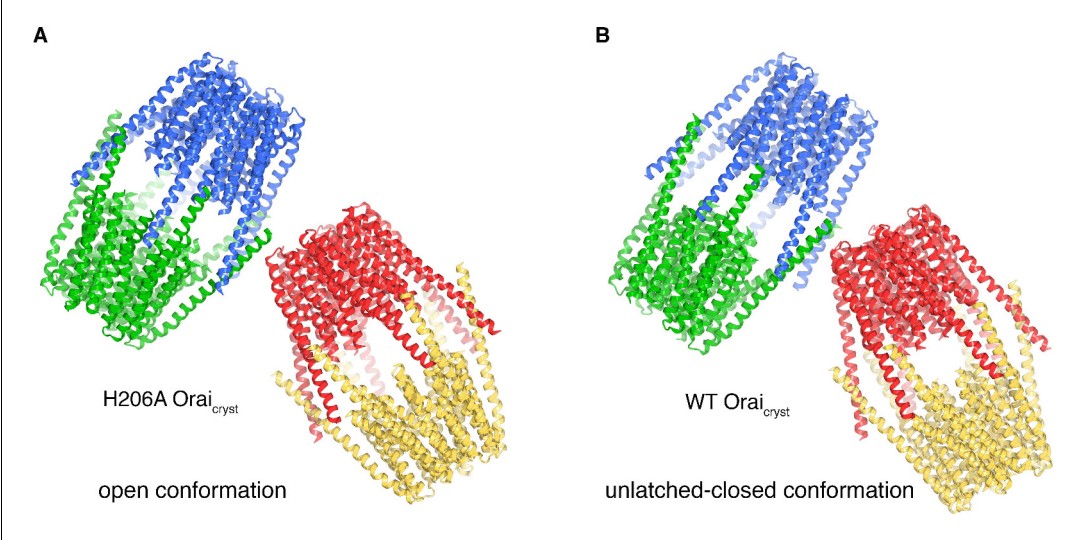

**Figure 8.** Molecular packing in the crystals of H206A Orai_cryst and WT Orai_cryst. (**a**) Crystal packing of H206A Orai_cryst. The contents of the asymmetric unit, consisting of four complete channels, is shown. Each channel is colored a unique color and shown in ribbon representation. The channels interact with one another via coiled-coil interactions between their M4-ext helices. (**b**) Packing of WT Orai_cryst in the crystal, showing the contents of the asymmetric unit, depicted analogously to **a**).

DOI: https://doi.org/10.7554/eLife.36758.015

The following figure supplement is available for figure 8:

**Figure supplement 1.** M4-ext helices and subunit comparisons.

DOI: https://doi.org/10.7554/eLife.36758.016

in a P4$_2$2$_1$2 crystal form that diffracted X-rays to 4.35 Å resolution, which assisted with model building and was indistinguishable from the I4$_1$ structures of WT and K163W Orai_cryst (*Figure 9—figure supplement 1*, and *Table 2*).

The structures of WT and K163W Orai_cryst reveal an 'unlatched-closed' conformation of the channel that resembles a hybrid of the quiescent and open conformations: the ion pores are closed as in the quiescent conformation but the M4/M4-ext regions have conformations like those observed in the H206A Orai_cryst structure (*Figure 9*). The unlatched-closed conformation has analogous straightenings of M4/M4-ext as the open structure, and analogous crystal packing (*Figure 8*). The conformation of the M1-M4a portion of the channel, however, is like the quiescent conformation (*Figure 9D*), and as such, the pores of WT and K163W Orai_cryst are closed and indistinguishable from the pore in the structure of the quiescent conformation (*Figure 9B,D*) (*Hou et al., 2012*). Therefore, contacts within the crystal and the straightening of M4/M4-ext are not responsible for the opening of the pore that is observed with H206A Orai_cryst.

As in the quiescent conformation, anomalous difference electron density for an iron complex is observed in the basic region of the pore in the unlatched-closed conformation (*Figure 9B*, *Figure 9—figure supplement 1C*). The density is positioned roughly in the center of the basic region in the structure of WT Orai_cryst and located in the lower portion of it in the structures of the K163W mutant, which removes the top ring of basic residues from the pore (*Figure 9B*, *Figure 9—figure supplement 1C*). The K163W mutation has no other apparent effect, and this is analogous to observations from structures of the quiescent conformation with and without the K163W mutation (*Hou et al., 2012*). The presence of the anomalous density in the basic region for both the quiescent and unlatched-closed conformations is further indication that the pores share the same closed conformation.

We wondered if the difference between the quiescent and unlatched-closed conformations could be due to subtle differences in protein constructs: mutations within the hypervariable M3-M4 loop (P276R and P277R) were made to obtain the well-ordered crystals that we used to determine the structure of the quiescent conformation (Orai_cryst-RR) but not used for the new structures. However, we also obtained crystals of Orai_cryst-RR in the same I4$_1$ crystal form and with nearly identical cell

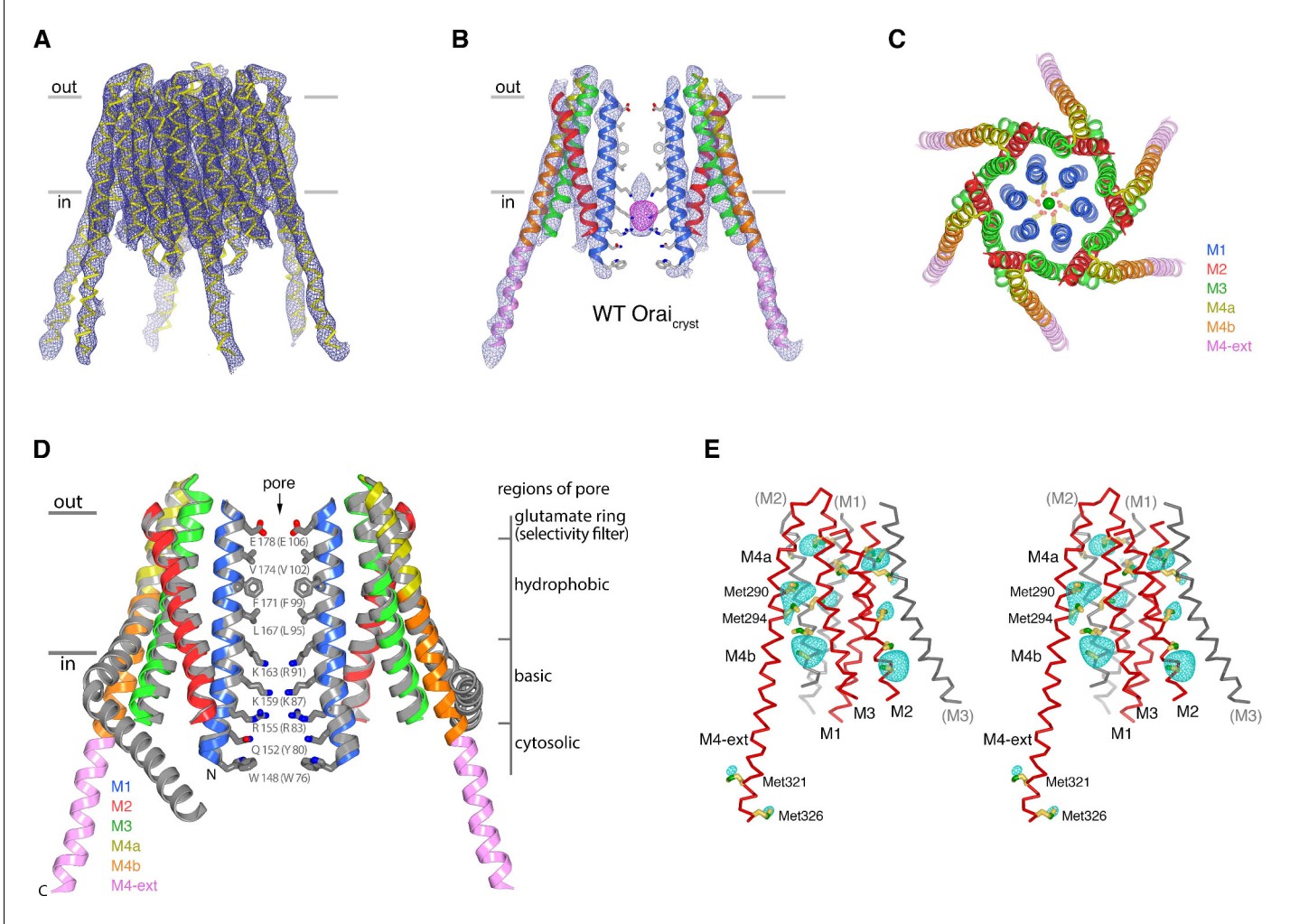

**Figure 9.** The structure of WT Orai_cryst reveals an unlatched-closed conformation. (a) Electron density for WT Orai_cryst, shown as blue mesh covering the channel (Cα representation). The map (contoured at 1.3 σ) was calculated from 20 to 6.9 Å using native sharpened amplitudes and phases that were determined by MR-SAD and improved by 24 fold NCS averaging, solvent flattening and histogram matching (Materials and methods). (b) Electron density, from (a) (blue mesh), covering two opposing subunits of WT Orai_cryst (cartoon representation, colored as indicated in c). Anomalous-difference electron-density (from iron) in the basic region of the pore is shown as magenta mesh (map calculated from 25 to 9 Å and contoured at 5 σ). Conformations of pore residues are based on the quiescent conformation (PDB ID 4HKR). (c) Extracellular view of WT Orai_cryst. Helices are drawn as ribbons and colored as indicated, with Glu178 side chains (sticks) and Ca²⁺ ion (green sphere) shown for reference. (d) Superposition of the quiescent conformation (PDB ID 4HKR) and the structure of the unlatched-closed conformation (WT Orai_cryst). Two subunits of each channel are shown. The quiescent conformation is gray; the structure of WT Orai_cryst is shown in colors. Amino acids lining the pore of the quiescent conformation are shown as sticks. A slight outward displacement of the intracellular side of M3 is observed in the structure of WT Orai_cryst; otherwise the conformations of M1-M4a are indistinguishable within the resolution limits of the diffraction data (RMSD for Cα positions 148 to 288 is 0.9 Å). (e) Anomalous-difference electron-density peaks at cysteine and methionine residues confirms the amino acid register of the WT Orai_cryst structure (stereo representation). An anomalous-difference electron-density map was calculated from 25 to 9 Å resolution from data collected with λ = 1.738 Å X-rays (Extended Data *Table 2*) using anomalous differences as amplitudes and phases from (a). This map was then averaged in real-space according to the 24-fold NCS symmetry to yield the map shown. The map is contoured at 5 σ (cyan mesh) and shown in the vicinity of a subunit of Orai (red Cα trace). Methionine and cysteine residues are shown as sticks (colored yellow for carbon and green for sulfur atoms). Methionine residues on M4b and M4-ext are labeled. Portions of neighboring Orai subunits (gray Cα traces) are shown for reference with their helices labeled in parentheses. Side chain conformations are hypothetical.
DOI: https://doi.org/10.7554/eLife.36758.017

The following figure supplement is available for figure 9:

**Figure supplement 1.** Structures of K163W Orai_cryst in I4₁ and P4₂2₁2 crystal forms (unlatched-closed conformation).
DOI: https://doi.org/10.7554/eLife.36758.018

dimensions as for the unlatched-closed conformation (unit cell dimensions: a = b = 265.5 Å, c = 224.2 Å, α=β=γ=90°). Although these crystals diffracted poorly (to ~9 Å resolution) and thus we did not pursue an X-ray structure, the correspondence of the crystal form indicates that Orai$_{cryst-RR}$ can also adopt a conformation with straightened M4/M4-ext helices. On the other hand, one might wonder if the P276R and P277R mutations could induce the pairing of the M4-ext helices observed what we refer to as the quiescent conformation. This possibility seems unlikely for the following four reasons: (1) Our data indicate that the Orai$_{cryst-RR}$ construct functions normally as a CRAC channel (*Figure 2*). (2) The amino acid substitutions were made to residues on the extracellular side of the channel, whereas the M4-ext helices are located on its intracellular side. (3) The M3-M4 loop varies dramatically among Orai channels both in terms of length and amino acid sequence. For example, the sequence of the corresponding region of human Orai1 is KQPGQPRPTSKPPASGAAANVSTSGI TPGQA (in single letter amino acid code; a sequence alignment can be found in *Hou et al. (2012)*). Furthermore, Orai from the deer tick *Ixodes scapularis* has an M3-M4 loop whose sequence is identical to *Drosophila* Orai except for Pro276 and Pro277, which are arginine and aspartate in that species, respectively, making it very similar to that of Orai$_{cryst-RR}$ (*Hou et al., 2012*). (4) The configuration of M4/M4-ext observed in the quiescent structure requires bends at the conserved Pro288 and SHK hinges. In support of a functional role for these bends, mutation of these hinge regions, which would presumably destabilize the quiescent conformation as discussed in more detail below, tend to activate the channel (*Nesin et al., 2014*; *Palty et al., 2015*; *Zhou et al., 2016*).

The X-ray structures of Orai suggest that an equilibrium exists between bent and unbent conformations of M4/M4-ext. Molecular constraints may bias this equilibrium during crystallization, and we hypothesize that STIM binding does so in a cellular context (as discussed later). In the absence of STIM, we suspect that fully bent (e.g. as observed in the quiescent structure) and fully straight (e.g. as observed in the unlatched-closed structure) conformations of M4/M4-ext are representative of a range of M4/M4-ext conformations that could be present in a cellular environment. Nevertheless, the ability of the M4 and M4-ext helices to adopt the configuration observed in the quiescent conformation, the sequence conservation of the hinges involved in the bends necessary to permit this conformation, and that the tendency of the M4-ext helices to pair is satisfied in the quiescent conformation, suggest physiological relevance for this conformation.

## Unlatching of M4b/M4-ext is necessary for pore opening

Comparison of the structures of the quiescent, unlatched-closed and open conformations indicates that the M4b and M4-ext regions must undergo conformational changes for the pore to open. In the quiescent conformation, the three sets of paired M4-ext helices create an assembly surrounding the intracellular side of the channel (*Figure 1A*). Bends at both Pro288 and the SHK motif are necessary for this configuration. As a result of the bend at Pro288 observed in the quiescent conformation, M4b interacts with M3 (*Figure 1A*, *Figure 10A*). In the open structure, the interaction between M4b and M3 is no longer present due to the repositioning of M4b that is enabled by unbending at Pro288 and the unpairing of M4-ext helices (*Figure 10C*). If the interaction between M4b and M3 of the quiescent conformation were present, or if the M4-ext helices were paired, the rigid body motion of M1-M4a that underlies pore opening could not occur due to steric interference between M3 and M4b (*Figure 10D*). We conclude that the paired M4-ext helices and the concomitant interactions between M4b and M3 of the quiescent conformation constitute 'latches' that must be released for the pore to open. In belt-like fashion, the latches constrain the outer diameter of the intracellular portion of the channel and prevent the widening observed in the H206A structure. Thus, when the latches are fastened, they stabilize the pore in a closed conformation. Complete straightenings of the M4/M4-ext helices (e.g. analogous to the conformations of the M4/M4-ext helices observed in the structures of H206A, WT, and K163W Orai$_{cryst}$) may not be necessary for the pore to open because there could be enough space for pore dilation with slight bends at the Pro288 and SHK hinges. Rather than forming rigid helices, we hypothesize that the Pro288 and SHK hinges provide flexibility to M4b and M4-ext when the latches are released. The straightened conformations of the M4/M4-ext helices in the crystal structures may represent only one conformation of these mobile regions along a continuum of unlatched conformations that would permit, and necessarily precede, the opening of the pore. The structures of the unlatched-closed conformation reveal that release of the latches does not necessarily open the pore: the pore is closed despite the M4b and M4-ext

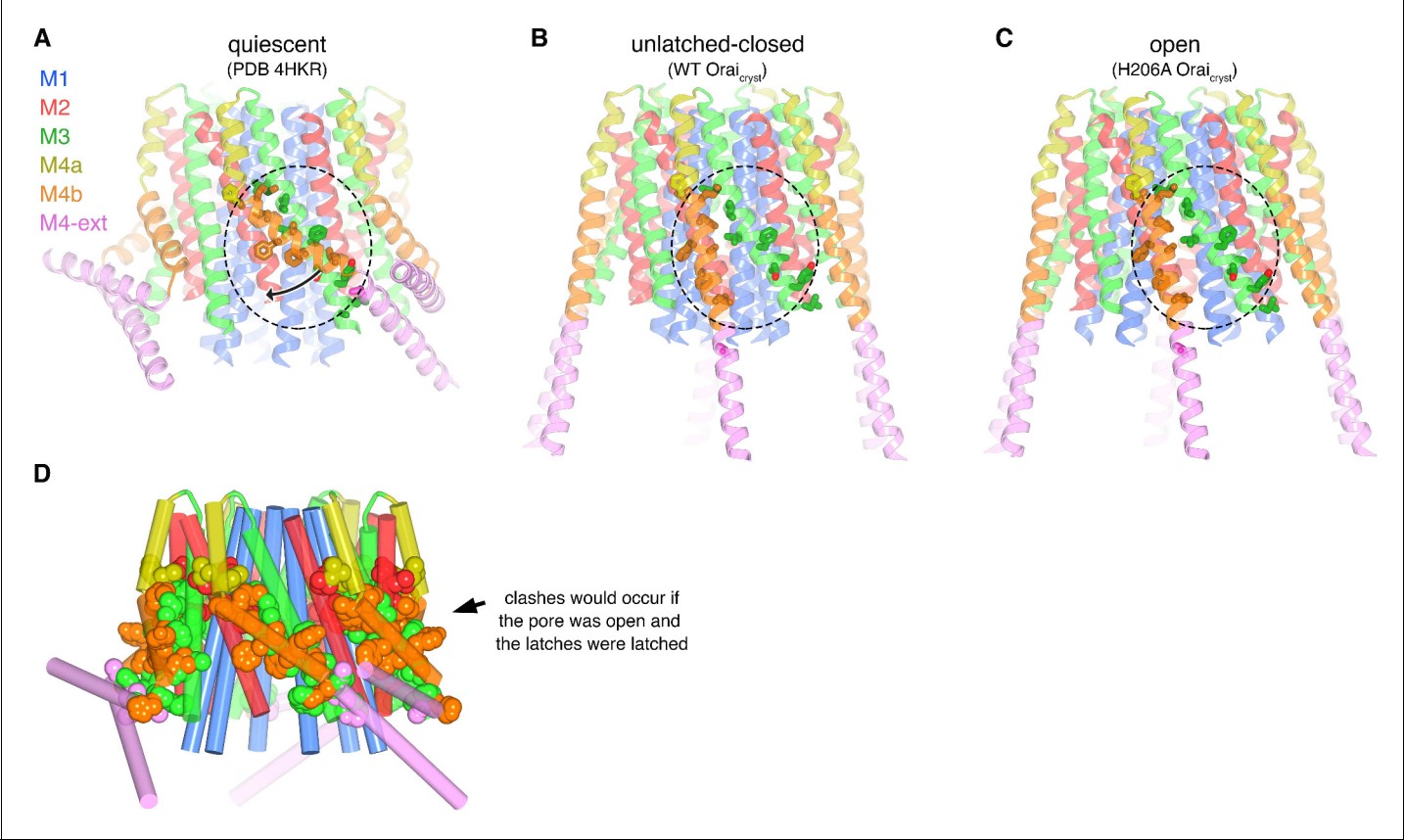

**Figure 10.** Unlatching is necessary for pore opening. (a) Quiescent conformation (PDB ID: 4HKR) highlighting interactions between M4b and M3. The channel is shown in cartoon representation. On one of the Orai subunits, amino acids in the interface between M4b (orange) and M3 (green), which is highlighted by a dashed oval, are shown as sticks and colored accordingly. This interface exists for all six subunits and is stabilized by the pairing of M4-ext helices (pink). An arrow denotes the movement of M4b between (a) and (b). (b) Conformation of WT Orai$_{cryst}$ showing the released latches and closed pore. The amino acids that had been in the interface between M4b and M3 in the quiescent conformation are drawn as sticks, with the dashed region showing the same region as in (a). (c) Open conformation (H206A Orai$_{cryst}$), depicted as in (b). (d) Unlatching is necessary for pore opening. A hybrid atomic model of the channel was generated using the conformations of the M1, M2 and M3 helices from the open conformation and the conformations of M4a/b and M4-ext from the quiescent conformation. In this model (shown in cartoon representation), molecular clashes exist between M4b and M3. Amino acids involved in the clashes are depicted as space-filling spheres (colored according to channel region: red, M2; green, M3; yellow, M4a; orange, M4b; pink, M4-ext). These steric hindrances would prevent opening of the pore while the latches are fastened. With unlatching, the repositioning of the M4b helix would allow the outward motion of M1-M3 that opens the pore.
DOI: https://doi.org/10.7554/eLife.36758.020

helices adopting the same conformation that they do in the H206A structure. Thus, while necessary, unlatching is not sufficient to open the pore.

In a cellular context, mutations that cause release of the latches, that is, those that destabilize the quiescent conformation and/or favor unbending at the Pro288 or SHK hinges, may appear as activating mutations. Congruently, Pro245 of human Orai1, which corresponds to Pro288 of Orai, has been characterized as a residue that helps stabilize a closed state of the channel - mutation to any other residue, which would favor straightening of the bend at this position, and thus destabilize a quiescent-like conformation, has an activating phenotype (*Nesin et al., 2014*; *Palty et al., 2015*). Further, certain mutations of human Orai1 within and around the SHK hinge also create active channels (*Zhou et al., 2016*). Because unlatching is necessary, but not sufficient, to open the pore, these mutations may increase the probability that the channel is open in the absence of STIM and/or they may increase the binding affinity for STIM. The drastic effects of mutations within the regions that we identify here as hinges suggest that properly functioning hinges are important for the conformational changes that underlie channel activation.

## Discussion

While the most pronounced structural differences between the closed and open pores are within the basic region, we find that removal of the basic region by mutation of these residues to serine does not form a constitutively open channel. On the other hand, mutations within the hydrophobic region of the pore (e.g. F99C or V102A of human Orai1 or V174A of Orai) give rise to leaky channels, albeit with diminished selectivity for $Ca^{2+}$ (*Figure 7*) (*Hou et al., 2012*; *McNally et al., 2012*; *Yamashita et al., 2017*). Although we cannot discern the conformations of amino acid side chains in the open structure, and thus there is a high degree of uncertainty when discussing the dimensions of the pore, the repositioning of the M1 helices in the opened pore indicates that the hydrophobic region widens markedly (*Figure 11*). We hypothesize that this widening, which accompanies the dramatic dilation of the basic region, is critical for ion permeation and that the hydrophobic region functions as a 'gate' – a variable constriction that prevents or permits ion conduction. Local conformational changes in the hydrophobic region and particularly at Phe171 (Phe99 in Orai1), such as changes in side chain rotamer conformations and/or the proposed slight rotation of M1 along its helical axis, which would move Phe171 further away from the central axis of pore (*Yamashita et al., 2017*), cannot be discerned from the X-ray diffraction data, but would be consistent with it, and these could contribute to pore widening and gating. High-resolution structural information of an opened pore, and preferably of an Orai-STIM complex, is necessary to discern the detailed physio-chemical environment along the pore that permits ion flow. Nevertheless, both structural and functional analyses point to the conclusion that while the conformational changes in the pore are more dramatic within the basic region, that they also extend through the hydrophobic region and to its extracellular side.

Ion binding in the open pore suggests that direct coordination of $Ca^{2+}$ by the ring of Glu178 residues in the selectivity filter is responsible for the channel's exquisite selectivity for $Ca^{2+}$. The proximity of the carboxylate moieties within the glutamate ring undoubtedly elevates the pKa values of the Glu178 side chains dramatically, such that they each would carry only a fraction of their −1 charge at physiological pH on average. (We estimated the pKa values of the Glu178 side chain carboxylates using a finite-difference Poisson-Boltzmann method, as implemented in the MCCE program

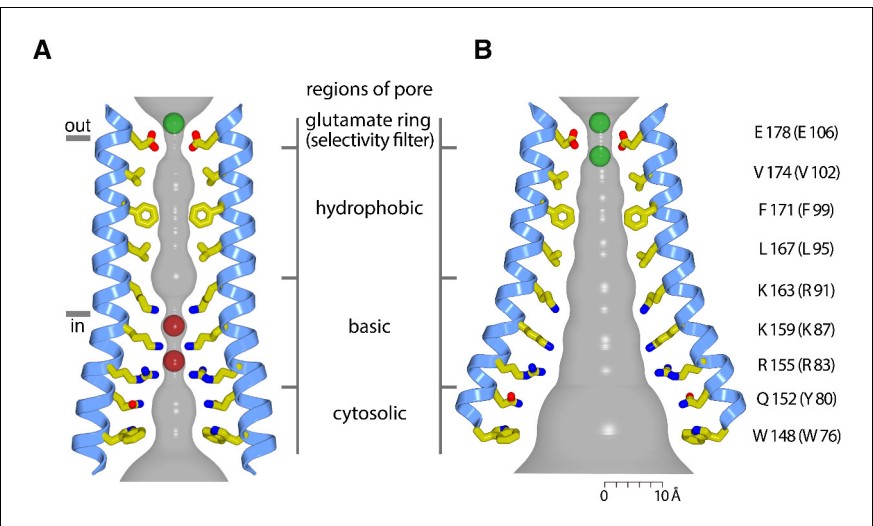

**Figure 11.** Hypothetical dimensions of the open pore. (a) The closed pore from the 3.35 Å resolution structure of the quiescent conformation of Orai (PDB 4HKR) (*Hou et al., 2012*). The gray surface represents the radial distance to the nearest van der Waals contact. $Ca^{2+}$ is represented as a green sphere. Red spheres represent the iron complex and anion binding sites in the closed pore. (b) Hypothetical dimensions of the open pore, which is depicted as in (a), based on the atomic coordinates of the H206A Orai_{cryst} structure. There is a large amount of uncertainty in the pore dimensions and in the conformations of side chains due to the limited resolution of the diffraction data. A scale bar is shown for reference. Two hypothetical $Ca^{2+}$ ions in the selectivity filter are depicted as green spheres.

DOI: https://doi.org/10.7554/eLife.36758.021

(*Georgescu et al., 2002*), and obtained values > 7, but caution that there is a large amount of uncertainty in these calculations. Nevertheless, it is well-established, in general, that amino acids in proteins can have dramatically altered ionization properties due the local environment of the surrounding protein compared with isolated amino acids in aqueous solutions [*Bashford and Karplus, 1990*]).

The structure provides context to the following hypothetical ion selectivity mechanism involving multiple ions in single file that is partially derived from what is known about ion selectivity mechanisms in Cav channels (*Sather and McCleskey, 2003*). In the physiological context of approximately 2 mM $Ca^{2+}$ outside the cell, we suspect that the selectivity filter of Orai toggles between having one and having two $Ca^{2+}$ ions present (although the two sites may have different free energies). When one $Ca^{2+}$ ion is bound, it would likely be centered within the glutamate ring, whereas when two are present, the ions would be positioned with one above the other (*Figure 11B*). The mutual repulsion between two $Ca^{2+}$ ions would allow either one to dissociate from the filter and when the lower one does it would begin to move through the pore. A single $Ca^{2+}$ ion remaining in the filter would not be easily displaced without the binding of a second $Ca^{2+}$ and thus its presence would block conduction of monovalent cations. Higher resolution structural information would undoubtedly provide additional insights into the mechanisms of ion selectivity in Orai.

Studies have shown that the M4-ext region of Orai interacts with a cytosolic portion of STIM that has a propensity to form coiled-coils (*Kawasaki et al., 2009*; *Li et al., 2007*; *Muik et al., 2008*; *Park et al., 2009*; *Yang et al., 2012*). Mutation of the residues corresponding to Ile316 and Leu319 in human Orai1, which mediate the coiled-coil packing in the crystal structures, prevents the interaction of Orai1 with STIM1 and subsequent channel activation (L273S and L276D mutations of human Orai1) (*Muik et al., 2008*; *Navarro-Borelly et al., 2008*). In accord with the unlatching we observe, an additional body of evidence suggests that channel activation involves conformational changes in the M4-ext helices (*Navarro-Borelly et al., 2008*; *Palty et al., 2017*; *Tirado-Lee et al., 2015*; *Zhou et al., 2016*). One possible mechanism of STIM binding is that unlatching would expose the M4-ext regions and make them available for interaction with STIM. Our observation that the unlatching of M4b from M3 does not necessarily open the pore is consistent with studies indicating that STIM1 can bind to loss-of-function mutants of human Orai1 that have constitutively closed pores, such as the pore-lining R91W mutant (*Derler et al., 2009*; *McNally et al., 2013*). Congruently, we observe that the corresponding K163W mutant of *Drosophila* Orai can adopt an unlatched conformation with a closed pore.

An important finding from our work is that the pore cannot open from the quiescent conformation directly. The paired M4-ext helices of the quiescent conformation necessitate the bend between M4a and M4b at Pro288 and they stabilize the interaction of M4b with M3 that would prevent the pore from opening. Therefore, the activation of Orai by STIM must involve different conformations of M4/M4-ext than those observed in the quiescent structure. On the other hand, the pore would be free to open once the latches are released. A logical hypothesis, which certainly requires further study, is that STIM activates the channel by binding to an unlatched conformation similar to what is observed in the unlatched-closed structure. An aspect of this hypothesis that is worth reiterating is that the M4 and M4-ext need not be perfectly straight for unlatching or for STIM binding.

Comparison of the three conformations of Orai revealed by the X-ray structures and the considerations mentioned above engender a hypothetical sequence for channel activation that proceeds from a quiescent state prior to interaction with STIM, through an unlatched conformation, and culminates with an open pore (*Figure 12*, *Figure 12—video 1*). In the quiescent conformation, clasped latches (e.g. the interactions between M4b and M3, which are stabilized by paired M4-ext helices) constrain the outer cytosolic diameter of the channel and prevent the pore from opening. Unlatching, that is, the release of the M4b-M3 interaction and the unpairing of M4-ext helices, could happen transiently and spontaneously, and would expose cytosolic docking sites for STIM. The engagement of STIM with Orai, via molecular interactions that remain to be resolved, would stabilize an unlatched conformation and the widening of the pore that permits $Ca^{2+}$ influx. The structures provide insight into the remarkable molecular choreography by which Orai governs store-operated $Ca^{2+}$ entry and a myriad of downstream cellular responses.

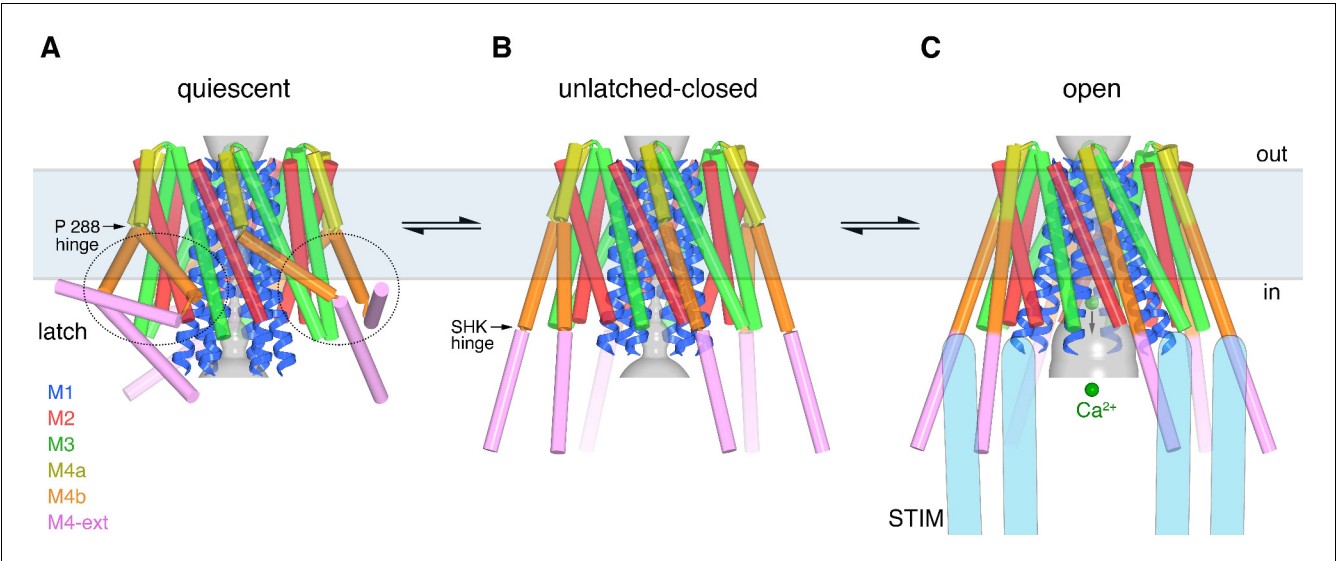

**Figure 12.** Proposed sequence of channel activation. (a) Quiescent conformation of Orai prior to binding of STIM (from PDB ID 4HKR, cylinders and ribbons). The pore (gray surface) is closed and the latches are fastened (two latches are indicated with dashed ovals). The M4 helices are bent at Pro288, delineating them into M4a and M4b. The M4b portions (orange) interact with the M3 helices (green), in six-fold fashion, and prevent the pore from opening by constraining the cytosolic region of the M3 helices. The interaction between M4b and M3 and the bend at Pro288 are stabilized by three sets of paired M4-ext helices. (b) An unlatched-closed conformation: structure of WT Orai$_{cryst}$ in which the pore is closed but the latches are released. Conformations of M1-M4a are indistinguishable from (a) (*Figure 8D*). When unlatched, mobile M4-ext regions are hypothesized to be available to interact with cytosolic regions of STIM that would become exposed as a result of depletion of Ca$^{2+}$ from the ER. Spontaneous unlatching would not necessarily require STIM binding and does not necessarily open the pore. (c) Open conformation. The structure of H206A Orai$_{cryst}$ is shown (cylinders and ribbons), with approximate dimensions of the pore shown as a gray surface. Following store depletion, we hypothesize that STIM (blue shapes) engages with cytosolic regions of Orai and stabilizes the pore in an open conformation. Unlatching is required to allow the widening of the pore and the influx of Ca$^{2+}$ (green spheres). Arrows between conformations denote equilibria and the horizontal rectangle indicates approximate boundaries of the plasma membrane. The depiction of the cytosolic region of STIM is conceptual and is not meant to imply stoichiometry or conformation.

DOI: https://doi.org/10.7554/eLife.36758.022

The following video is available for figure 12:

**Figure 12—video 1.** Video showing hypothetical opening sequence as illustrated in *Figure 12*.

DOI: https://doi.org/10.7554/eLife.36758.023

# Materials and methods

## Cloning, expression, purification and crystallization

cDNA encoding *Drosophila melanogaster* Orai (amino acids 133–341) followed by a C-terminal YL½ antibody affinity tag (amino acids EGEEF)(*Kilmartin et al., 1982*) was cloned into the EcoRI and NotI restriction sites of the *Pichia pastoris* expression vector pPICZ-C (Invitrogen Life Technologies). Two non-conserved cysteine residues were mutated to improve protein stability (C224S and C283T). This construct, termed 'WT Orai$_{cryst}$', differs from the one we used previously (*Hou et al., 2012*) only in that it contains wild-type Pro276 and Pro277 residues in the hypervariable M3-M4 loop rather than arginine substitutions at these positions. Constructs bearing the H206A or K163W mutations were made on the background of WT Orai$_{cryst}$ using standard molecular biology techniques (designated 'H206A Orai$_{cryst}$' or 'K163W Orai$_{cryst}$', accordingly). Transformations into *P. pastoris*, expression, and cell lysis were performed as previously described (*Long et al., 2005*).

Lysed *P. pastoris* cells were re-suspended in buffer (3.3 ml buffer for each 1 g of cells) containing 150 mM KCl, 10 mM sodium phosphate, pH 7.0, 0.1 mg/ml deoxyribonuclease I (Sigma-Aldrich), 1:1000 dilution of Protease Inhibitor Cocktail Set III, EDTA free (CalBiochem), 1 mM benzamidine (Sigma-Aldrich), 0.5 mM 4-(2-aminoethyl) benzenesulfonyl fluoride hydrochloride (Gold Biotechnology) and 0.1 mg/ml soybean trypsin inhibitor (Sigma-Aldrich). Cell lysate was adjusted to pH 7.0 with 1 N KOH, 0.11 g n-dodecyl-β-D-maltopyranoside (DDM, Anatrace, solgrade) per 1 g of cells

**Table 1.** H206A Orai$_{cryst}$ data collection, phasing and refinement statistics.

Data collection statistics are from HKL3000 (*Otwinowski and Minor, 1997*) or XDS (I$^-$ experiment) (*Kabsch, 2010*). $R_{sym} = \Sigma \mid I_i - < I_i > \mid / \Sigma \mid I_i$, where $< I_i >$ is the average intensity of symmetry-equivalent reflections. $CC_{1/2}$, $CC_{work}$ and $CC_{free}$ are defined in (*Karplus and Diederichs, 2012*). Phasing power = RMS ($\mid F \mid / \varepsilon$), where $\mid F \mid$ is the heavy-atom structure factor amplitude and $\varepsilon$ is the residual lack of closure error. $R_{cullis}$ is the mean residual lack of closure error divided by the dispersive or anomalous difference. $R_{work} = \Sigma \mid F_{obs} - F_{calc} \mid / \Sigma \mid F_{obs} \mid$, where $F_{obs}$ and $F_{calc}$ are the observed and calculated structure factors, respectively. $R_{free}$ is calculated using a subset (~10%) of reflection data chosen randomly and omitted throughout refinement. Figure of merit is indicated after density modification and phase extension starting from 9.0 Å in DM. R.m.s.d: root mean square deviations from ideal geometry. Numbers in parentheses indicate the highest resolution shells and their statistics.

| | H206A Orai$_{cryst}$ | | | |
| --- | --- | --- | --- | --- |
| | **Native** | **Ba$^{2+}$** | **Gd$^{3+}$** | **I$^-$** |
| Space group | I4$_1$ | I4$_1$ | I4$_1$ | I4$_1$ |
| Datasets source | APS 24ID-C | APS 24ID-C | APS 23ID-D | APS 23ID-D |
| Wavelength (Å) | 1.1000 | 1.7000 | 1.7000 | 1.7085 |
| Cell dimensions: | | | | |
| a, b, c (Å) | 262.3, 262.3, 220.4 | 265.0, 265.0, 219.9 | 255.4, 255.4, 216.0 | 266.4, 266.4, 221.5 |
| α= β= γ (°) | 90 | 90 | 90 | 90 |
| Resolution (Å) | 50–6.70 (6.82–6.70) | 50–7.40 (7.53–7.40) | 50–7.90 (8.04–7.90) | 50–7.6 (7.68–7.60) |
| No. of crystals | 1 | 1 | 1 | 2 |
| R$_{sym}$ (%) | 8.3 (>100) | 22.5 (>100) | 12.1 (>100) | 14.0 (>100) |
| R$_{pim}$ (%) | 1.1 (57.0) | 3.4 (46.7) | 2.4 (42.8) | 1.1 (62.0) |
| CC$_{1/2}$ (in outer shell) | 0.214 | 0.173 | 0.158 | 0.493 |
| I/σI | 92.4 (1.1) | 47.0 (1.4) | 37.5 (2.0) | 26.8 (0.2) |
| Completeness (%) | 100.0 (100.0) | 100.0 (100.0) | 100.0 (100.0) | 99.9 (99.7) |
| Redundancy | 55.8 (58.7) | 46.5 (51.8) | 25.1 (26.0) | 165.1 (172.5) |
| Figure of Merit (DM) | 0.729 [20–6.7 Å] | | | |
| Refinement | PDB ID: 6BBF | | | |
| Resolution (Å) | 20–6.7 | | | |
| No. of reflections | 12844 | | | |
| R$_{work}$ (%) | 30.6 | | | |
| R$_{free}$ (%) | 33.9 | | | |
| CC$_{work}$/CC$_{free}$ (in outer shell) | 0.285/0.220 | | | |
| No. atoms | 27120 | | | |
| Ramachandran (%) | | | | |
| Favored | 97.22 | | | |
| Outliers | 1.04 | | | |
| R.m.s.d: | | | | |
| bond lengths (Å) | 0.005 | | | |
| bond angles (°) | 1.15 | | | |

DOI: https://doi.org/10.7554/eLife.36758.008

was added to the cell lysate, and the mixture was stirred at room temperature for 45 min to extract Orai from the membranes. The sample was then centrifuged at 30,000 *g* for 45 min at 17°C and the supernatant was filtered (0.45 µm polyethersulfone membrane). YL½ antibody (IgG, expressed from hybridoma cells and purified by ion exchange chromatography) was coupled to CNBr-activated sepharose beads (GE Healthcare) according to the manufacturer's protocol. Approximately 0.4 ml of beads were added to the sample for each 1 g of *P. pastoris* cells and the mixture was rotated at room temperature for 1 hr. Beads were collected on a column, washed with five column-volumes of buffer containing 150 mM KCl, 10 mM sodium phosphate, pH 7.0, 5 mM DDM, 0.1 mg/ml lipids

**Table 2.** Data collection, phasing and refinement statistics for WT and K163W Orai$_{cryst}$.

Data collection statistics are from HKL3000 (*Otwinowski and Minor, 1997*). R$_{sym}$ = $\Sigma$ | I$_i$- < I$_i$ > | / $\Sigma$ I$_i$, where < I$_i$ > is the average intensity of symmetry-equivalent reflections. CC$_{1/2}$, CC$_{work}$ and CC$_{free}$ are defined in (*Karplus and Diederichs, 2012*). Phasing power = RMS (|F|/$\varepsilon$), where |F| is the heavy-atom structure factor amplitude and $\varepsilon$ is the residual lack of closure error. R$_{cullis}$ is the mean residual lack of closure error divided by the dispersive or anomalous difference. R$_{work}$ = $\Sigma$ | F$_{obs}$ − F$_{calc}$ | / $\Sigma$ | F$_{obs}$ |, where F$_{obs}$ and F$_{calc}$ are the observed and calculated structure factors, respectively. R$_{free}$ is calculated using a subset (~10%) of reflection data chosen randomly and omitted throughout refinement. Figure of merit is indicated after density modification and phase extension starting from 8.0 Å in DM. R.m.s.d: root mean square deviations from ideal geometry. Numbers in parentheses indicate the highest resolution shells and their statistics.

| | K163W Orai$_{cryst}$ | K163W Orai$_{cryst}$ | | | WT Orai$_{cryst}$ |
| | **Native** | **Native** | **Derivative 1**<br>**PCMB** | **Derivative 2**<br>**PIP** | **Native** |
|---|---|---|---|---|---|
| Space group | P4$_2$2$_1$2 | I4$_1$ | I4$_1$ | I4$_1$ | I4$_1$ |
| Datasets source | NSLS X25 | NSLS X25 | NSLS X29 | NSLS X29 | NSLS X25 |
| Wavelength (Å) | 1.1000 | 1.1000 | 1.0074 | 1.0712 | 1.738 |
| Cell dimensions: | | | | | |
| a, b, c (Å) | 118.7, 118.7, 122.4 | 247.5, 247.5, 210.2 | 246.1, 246.1, 210.0 | 250.6, 250.6, 211.8 | 250.4, 250.4, 210.4 |
| α= β= γ (°) | 90 | 90 | 90 | 90 | 90 |
| Resolution (Å) | 60–4.35 (4.42–4.35) | 50–6.10 (6.20–6.10) | 50–6.10 (6.20–6.10) | 50–6.90 (7.02–6.90) | 50–6.9 (7.02–6.90) |
| No. of crystals | 1 | 1 | 1 | 1 | 1 |
| R$_{sym}$ (%) | 6.0 (>100) | 5.9 (>100) | 5.7 (>100) | 12.0 (>100) | 9.4 (>100) |
| R$_{pim}$ (%) | 1.7 (55.7) | 1.3 (>100) | 1.9 (>100) | 3.1 (>100) | 3.0 (>100) |
| CC$_{1/2}$ (in outer shell) | 0.265 | 0.376 | 0.378 | 0.170 | 0.123 |
| I/σI | 61.1 (1.0) | 69.0 (1.0) | 49.3 (0.7) | 37.0 (.07) | 39.1 (0.5) |
| Completeness (%) | 100.0 (100.0) | 100.0 (100.0) | 99.9 (100) | 99.9 (100) | 99.7 (99.8) |
| Redundancy | 16.4 (16.9) | 22.9 (23.9) | 21.5 (20.5) | 16.3 (16.7) | 11.0 (11.6) |
| MIRAS Phasing | | | | | |
| No. of sites | | | 24 | 24 | |
| Phasing power (iso/ano) | | | 0.523/0.559 | 0.405/0.686 | |
| R$_{cullis}$ (iso/ano) | | | 0.764/0.945 | 0.952/0.919 | |
| Figure of Merit (DM) | 0.627 [20–4.35 Å] | 0.629 [20–6.1 Å] | | | 0.777 [20–6.9 Å] |
| Refinement | PDB ID: 6BBI | PDB ID: 6BBH | | | PDB ID: 6BBG |
| Resolution (Å) | 20–4.35 | 20–6.1 | | | 20–6.9 |
| No. of reflections | 6035 | 14682 | | | 10287 |
| R$_{work}$ (%) | 30.6 | 31.4 | | | 33.4 |
| R$_{free}$ (%) | 32.9 | 34.0 | | | 35.4 |
| CC$_{work}$/CC$_{free}$ (in outer shell) | 0.487/0.441 | 0.346/0.333 | | | 0.332/0.414 |
| No. atoms | 3338 | 27360 | | | 27240 |
| Ramachandran (%) | | | | | |
| Favored | 95.5 | 97.2 | | | 96.6 |
| Outliers | 0.24 | 1.01 | | | 0.93 |
| R.m.s.d: | | | | | |
| Bond lengths (Å) | 0.006 | 0.005 | | | 0.005 |
| Bond angles (°) | 1.15 | 1.06 | | | 1.07 |

DOI: https://doi.org/10.7554/eLife.36758.019

(3:1:1 molar ratio of 1-palmitoyl-2-oleoyl-sn-glycero-3-phosphocholine, 1-Palmitoyl-2-oleoyl-sn-glycero-3-phosphoethanolamine, and 1-palmitoyl-2-oleoyl-sn-glycero-3-[phospho-rac-(1-glycerol)], obtained from Avanti) and eluted with buffer containing 150 mM KCl, 100 mM Tris-HCl, pH 8.5, 5 mM DDM, 0.1 mg/ml lipids and 5 mM Asp-Phe peptide (Sigma-Aldrich). The eluted protein was concentrated to ~25 mg/ml using a 100 kDa concentrator (Amicon Ultra, Millipore) and further purified on a Superdex-200 gel filtration column (GE Healthcare) in 75 mM KCl, 10 mM Tris-HCl, pH 8.5, 0.1 mg/ml lipids, and detergent: 4 mM octyl glucose neopentyl glycol (Anatrace, anagrade) to obtain crystals of K163W Orai$_{cryst}$ in space group P4$_2$2$_1$2, or a mixture of 0.5 mM decyl maltose neopentyl glycol (Anatrace, anagrade) and 3 mM octyl glucose neopentyl glycol for crystals of WT and K163W Orai$_{cryst}$ in space group of I4$_1$, and 0.5 mM decyl maltose neopentyl glycol for H206A Orai$_{cryst}$. For crystals of H206A Orai$_{cryst}$, 3 mM octyl glucose neopentyl glycol was added into the purified H206A Orai$_{cryst}$ just before setting up crystallization trials. A typical prep, utilizing 20 g of cells, yielded ~2 mg of purified Orai. For the crystals of H206A Orai$_{cryst}$ with I$^-$, NaI was substituted in place of KCl in the purification buffers. For crystals of H206A in Ba$^{2+}$, 5 mM BaCl$_2$ was added to the final purified protein before crystallization. Purified Orai proteins were concentrated to 10–20 mg/ml using 100 kDa Vivaspin-2 concentrators (Sartorius Stedim Biotech), and mixed 1:1 (250 nl: 250 nl) with crystallization solutions for hanging drop vapor diffusion crystallization. Crystals of WT Orai$_{cryst}$ grew in 32–35% PEG 400 (v/v) and 0.1 M potassium phosphate pH 7.5. Crystals of K163W Orai$_{cryst}$ in space group P4$_2$2$_1$2 grew in 24–26% PEG 400 (v/v) and 0.2 M potassium phosphate pH 6.5. Crystals of K163W Orai$_{cryst}$ in space group I4$_1$ grew in 26–28% PEG 400 (v/v) and 150 mM NaCl and 100 mM N-2-hydroxyethylpiperazine-N-2'-ethanesulfonic acid (HEPES) pH 7.5. Native crystals of H206A Orai$_{cryst}$ grew in 36–38% PEG400, 500 mM NaCl and 100 mM Tris-HCl pH 9.0. The crystallization solution for H206A Orai$_{cryst}$ with NaI was 28–31% PEG400 and 100 mM Tris-HCl pH7.5. The crystallization solution for H206A Orai$_{cryst}$ with BaCl$_2$ was 32–34% PEG400, 500 mM NaCl and 100 mM Tris-HCl pH 8.5.

## Structure determination

### Data collection and processing

All crystals were dehydrated and cryo-protected before flash-cooling in liquid nitrogen by serial transfer into solutions containing the buffer components of an equilibrated crystallization drop and increasing concentrations of PEG 400 (to 50% w/v) in nine steps with ~1 min intervals. For heavy atom derivatives, Crystals of K163W Orai$_{cryst}$ belonging to space group I4$_1$ were soaked in stabilization solution supplemented with ~18 µg/ml p-chloromercuribenzene sulfate (PCMB), or ~7 µg/ml di-µ-iodo-bis(ethylene-diamine)-di-platinum(II) nitrate (PIP) for 24 hr. For ion binding experiments, crystals of H206A Orai$_{cryst}$ were soaked in stabilization solution supplemented with 1 mM GdCl$_3$ for 2 days. After soaking, the crystals were cryo-protected in the same solutions as native crystals and flash-cooled. Crystals of H206A Orai$_{cryst}$ that contained 5 mM BaCl$_2$ were soaked in stabilization solution supplemented with 50 mM BaCl$_2$ during dehydration steps and flash-cooled. X-ray diffraction data sets were collected using synchrotron radiation and were indexed, integrated and scaled with the HKL suite (Otwinowski & Minor, 1997) or XDS (Kabsch, 2010). Resolution limits of the diffraction data were estimated from the CC$_{1/2}$ value (*Karplus and Diederichs, 2012*).

### K163W Orai$_{cryst}$ (P4$_2$2$_1$2 space group)

Initial phases for data collected from crystals of K163W Orai$_{cryst}$ belonging to space group P4$_2$2$_1$2, were determined by molecular replacement (MR) with PHENIX (*Adams et al., 2010*) using residues 148–288 of the structure of K163W *Drosophila* Orai in the quiescent conformation (PDB ID: 4HKS) as a search model. The asymmetric unit contains three Orai subunits; these form a complete hexameric channel by a two-fold rotational symmetry operator of the P4$_2$2$_1$2 space group. To improve the phases and reduce bias, the phases were improved with solvent flattening, histogram matching, and 3-fold non-crystallographic symmetry (NCS) averaging with the program DM (*Cowtan, 1994*). This yielded well-defined density for the channel (*Figure 9—figure supplement 1*). A B-factor sharpening value of −150 Å$^2$ was applied to the electron density maps that are displayed (*Figure 4*, *Figure 9*, and *Figure 9—figure supplement 1*). The atomic model was adjusted in COOT (*Emsley et al., 2010*) and refined in CNS using a deformable elastic network (DEN) force field (*Brünger et al., 1998*; *Brunger et al., 2012*; *Schröder et al., 2007*; *Schröder et al., 2010*) and in PHENIX with NCS

and secondary structure restraints. The final model contains residues 148–327 of Orai, excluding the following residues that did not have well-enough defined electron density to direct model building: 181–188 (the M1-M2 loop), 220–239 (the M2-M3 loop), and 314–327 of subunit B.

## K163W Orai$_{cryst}$ (I4$_1$ space group)

Initial phases for K163W Orai$_{cryst}$ in space group I4$_1$ were determined experimentally by the MIRAS (multiple isomorphous replacement with anomalous scattering) method using a native dataset and two derivative ones (PCMB and PIP) using SHARP (*Vonrhein et al., 2007*) (*Table 2*; *Figure 9—figure supplement 1E*). The asymmetric unit contains four hexameric channels (24 Orai subunits) for which density was apparent following solvent flattening with the program DM (*Cowtan, 1994*). The phases were improved and extended to 6.1 Å resolution using solvent flattening, histogram matching, and 24-fold NCS averaging with DM. This yielded continuous densities for all helices in all 24 Orai subunits within the asymmetric unit (*Figure 9—figure supplement 1A*). Anomalous-difference electron density maps of well-ordered platinum and mercury sites at Met321 and Cys215, respectively, helped establish the amino acid register. The quiescent conformation structure (PDB ID: 4HKS; determined at 3.35 Å) was used as a reference for modeling of the M1-M3 portion and the 4.35 Å structure in space group P4$_2$2$_1$2 was used for M4/M4-ext. Minor adjustments of the M4 and M4-ext helices were made as necessary. Refinement was done using rigid body and DEN refinement in CNS utilizing NCS, helical secondary structure (backbone phi, psi) and phase restraints (*Brünger et al., 1998*; *Brunger et al., 2012*; *Schröder et al., 2007*; *Schröder et al., 2010*). Grouped B-factor and TLS refinement were performed (in PHENIX), for which each of the four channels was defined as a group, as is appropriate for modest-resolution data. The final model contains residues 148–327 of Orai, excluding the following residues that did not have well-enough defined electron density to direct model building: 181–188 (the M1-M2 loop) and 220–239 (the M2-M3 loop).

## WT Orai$_{cryst}$ (I4$_1$ space group)

Initial phases for the structure of WT Orai$_{cryst}$ were determined by MR using the K163W Orai$_{cryst}$ structure (space group I4$_1$) as an initial model in the program PHENIX (*Adams et al., 2010*). Diffraction data were collected to maximize the anomalous scattering from iron ($\lambda$ = 1.738 Å). Single-wavelength anomalous diffraction (SAD) phases derived from the anomalous density (presumably from iron) in the basic region of the pore were combined with the molecular replacement phases (MR-SAD method) using AutoSol of PHENIX (*Adams et al., 2010*). Potential phase bias was further minimized by density modification with solvent flattening, histogram matching, and 24-fold NCS averaging using the program DM (*Cowtan, 1994*). Model building was aided by the quiescent conformation (PDB ID: 4HKR; 3.35 Å resolution) and by the 4.35 Å resolution structure of K163W Orai$_{cryst}$ from the P4$_2$2$_1$2 space group. Minor adjustments of the M4-ext helices were made in COOT (*Emsley et al., 2010*). Refinement was done using rigid body and DEN refinement in CNS (*Brünger et al., 1998*; *Brunger et al., 2012*; *Schröder et al., 2007*; *Schröder et al., 2010*). During refinement, NCS, helical secondary structure (backbone phi, psi) restraints and experimental phase restraints were applied. Grouped B-factor and TLS refinement were performed (in PHENIX), for which each of the four channels was defined as a group. The final model contains residues 148–327 of Orai, excluding the following residues that did not have well-enough defined electron density to direct model building: 181–191 (the M1-M2 loop) and 220–239 (the M2-M3 loop).

## H206A Orai$_{cryst}$ (I4$_1$ space group)

Initial phases for the structure of H206A Orai$_{cryst}$ were obtained with MR using a truncated structure (amino acids 148–309) of WT Orai$_{cryst}$ (I4$_1$ space group) as a search model in PHENIX. At this stage, electron density maps contained broken density for the four channels in the asymmetric unit. These phases were used to identify four Gd$^{3+}$ sites (one site in the glutamate ring of each channel in the ASU) from the dataset collected from a crystal soaked in GdCl$_3$ (*Table 1*) and the phases were improved using the MR-SAD method (using AutoSol in PHENIX). The phases were then improved using solvent flattening, histogram matching, and four-fold non-crystallographic symmetry (NCS) averaging with the program DM (*Cowtan, 1994*) using an entire channel as the reference region for NCS averaging. This yielded continuous electron density for helices. These phases were used as starting phases for the native dataset and were improved and extended to 6.7 Å resolution using

the 24-fold NCS present within the asymmetric unit. For the 24-fold averaging, a single Orai subunit corresponding to amino acids 148–309 was used as the reference region (with solvent flattening, histogram matching, and NCS averaging performed in DM). This map (shown in *Figure 4A,B*) was used to direct model building. The initial model was generated by rigid body fit of WT Orai$_{cryst}$ subunits and adjusted manually in COOT (e.g. to account for the additional bend in M1) (*Emsley et al., 2010*). Side chain conformations cannot be determined from the electron density due to the limit of the diffraction data; side chains are included in the atomic model for reference, however, and their conformations are based on those observed in the quiescent conformation (PDB 4HKR) for amino acids in M1-M4a and those from the 4.35 Å resolution structure presented here (P4$_2$2$_1$2 space group of K163W Orai$_{cryst}$) for amino acids in M4b and M4-ext. Helical regions were modeled with ideal α-helical geometry and side chain rotamers were selected from frequently occurring conformations (*Hintze et al., 2016*) and to minimize steric clashes (*Word et al., 1999*). Refinement was done using rigid body and DEN refinement in CNS utilizing NCS and helical secondary structure (backbone phi, psi) restraints (*Brünger et al., 1998*; *Brunger et al., 2012*; *Schröder et al., 2007*; *Schröder et al., 2010*). Grouped B-factor and TLS refinement were performed (in PHENIX), for which each of the four channels was defined as a group. Highly redundant data allowed us to visualize anomalous-difference electron density arising from the sulfur atoms of methionine or cysteine residues on each of the M1-M4 helices and on the M4-ext helix (*Table 1*). These anomalous peaks confirm the assigned amino acid register and indicate the accuracy of the crystallographic phases (*Figure 4—figure supplement 1*). The model contains residues 148–327 of Orai, excluding the following residues that did not have well-enough defined electron density to direct model building: 181–191 (the M1-M2 loop) and 220–239 (the M2-M3 loop).

### Anomalous-difference electron density maps for ion experiments

Phases for the three anomalous-difference electron density maps in *Figure 6* (*Table 1*) were determined by MR-SAD (Phenix, using H206A Orai$_{cryst}$ for MR and the anomalous signal from ions), and were improved by 24-fold NCS averaging, solvent flattening and histogram matching in DM. Anomalous difference electron density for each ion was observed, with approximately the same sigma level and position, in all four channels of each asymmetric unit.

### Reconstitution and flux assay

Orai constructs were purified and reconstituted into lipid vesicles using a modified published procedure (*Hou et al., 2012*). A lipid mixture containing 15 mg/ml POPE (1-palmitoyl-2-oleoyl-sn-glycero-3-phosphocholine) and 5 mg/ml POPG (1-palmitoyl-2-oleoyl-sn-glycero-3-phospho(1'-rac-glycerol)) was prepared in water and solubilized with 8% (w/vol) n-decyl-β-D-maltopyranoside. Purified WT or H206A Orai$_{cryst}$ protein was mixed with the solubilized lipids to obtain a final protein concentration of 0.5 mg/ml and a lipid concentration of 10 mg/ml. Detergent was removed by dialysis (15 kDa molecular weight cutoff) at 4°C for 7 days against a reconstitution buffer containing 10 mM HEPES pH 7.0, 150 mM KCl and 0.2 mM ethylene glycol tetraacetic acid (EGTA), with daily buffer exchanges and utilizing a total volume of 14 l of reconstitution buffer. The reconstituted sample was sonicated (~30 s), aliquoted, flash-frozen in liquid nitrogen and stored at −80°C.

Vesicles were rapidly thawed (using 37°C water bath), sonicated for 5 s, incubated at room temperature for 2–4 hr before use, and then diluted 100-fold into a flux assay buffer containing 150 mM n-methyl-d-glucamine (NMDG), 10 mM HEPES pH 7.0, 0.2 mM EGTA, 0.5 mg/mL bovine serum albumin and 0.2 μM 9-amino-6-chloro-2-methoxyacridine (ACMA, from a 2 mM stock in DMSO). Data were collected on a SpectraMax M5 fluorometer (Molecular Devices) using Softmax Pro five software. Fluorescence intensity measurements were collected every 30 s over the span of the 1200 s experiment (excitation and emission set to 410 nm and 490 nm, respectively). The proton ionophore carbonyl cyanide m-chlorophenyl hydrazine (CCCP, 1 μM from a 1 mM stock in DMSO) was added after 150 s and the sample was mixed briefly by pipette. The potassium ionophore valinomycin (2 nM from a 2 uM stock in DMSO) was added at the end of the experiment (990 s) to establish a baseline fluorescence and confirm vesicle integrity. For experiments used to determine the effects of Ca$^{2+}$, Mg$^{2+}$ or Gd$^{3+}$ on H206A Orai$_{cryst}$, CaCl$_2$ (1 mM final concentration), MgCl$_2$ (3 mM final concentration), or GdCl$_3$ (0.1 mM final concentration) was added to aliquots of reconstituted vesicles. To introduce the ions into the vesicles, these vesicles were sonicated for 5 s, frozen in liquid

nitrogen, thawed, sonicated a second time for 5 s, and incubated at room temperature for 2–4 hr prior to measurements. The flux assay buffers were supplemented with 1 mM $CaCl_2$, 3 mM $MgCl_2$, or 0.1 mM $GdCl_3$, respectively, for these experiments.

Flux experiments using V174A Orai$_{cryst}$ and Orai bearing the simultaneous R155S, K159S, and K163S mutations (SSS Orai$_{cryst}$) were performed analogously. In these experiments, $Na^+$ flux was measured under divalent-free conditions as described (*Hou et al., 2012*). Purified protein was prepared as described above. Proteoliposomes, or liposomes without protein, were formed by dialysis against 10 mM HEPES, pH 7.0, 150 mM NaCl and 0.2 mM EGTA, and were aliquoted, flash-frozen in liquid nitrogen and stored at −80°C until use. For reconstitution, the lipid concentration was 10 mg/ml POPE:POPG (3:1 wt ratio) and the protein concentrations were 1 mg/ml and 0.01 mg/ml for SSS Orai$_{cryst}$ and V174A Orai$_{cryst}$, respectively. (0.1 and 0.01 mg/ml SSS Orai$_{cryst}$ concentrations were also tested and gave indistinguishable results.) Liposome samples were diluted 50-fold into flux buffer containing 10 mM HEPES pH 7.0, 0.2 mM EGTA, 0.5 mg/mL bovine serum albumin, 0.2 μM ACMA, and 150 mM N-methyl-D-glucamine (NMDG), which established a $Na^+$ gradient. After stabilization of the fluorescence signal (150 s), 1 μM CCCP was added to the sample. The $Na^+$ ionophore monensin was added after 990 s to render all vesicles permeable to $Na^+$ and establish the minimum baseline fluorescence.

Incorporation of purified Orai proteins into proteoliposomes was assessed by solubilizing proteoliposomes containing 0.1 mg/ml Orai$_{cyst}$ or SSS Orai$_{cryst}$ in buffer containing 40 mM Tris-HCl pH8.5, 150 mM NaCl and 10 mM DDM at 4°C for 1 hr. After the samples were clarified by centrifugation at 20,800 *g* for 1 hr at 4°C before SEC analysis. Supernatants were loaded onto a Superose 6 Increase column (10/300 GL; GE healthcare) in buffer containing 10 mM Tris-HCl pH8.5, 150 mM NaCl and 1 mM DDM, using tryptophan fluorescence for protein detection.

## Live cell $Ca^{2+}$ influx measurements with GCaMP6s

Orai constructs were expressed with N-terminal mCherry tags using a modified pNGFP-EU vector (*Kawate and Gouaux, 2006*) in which cDNA encoding GFP was replaced with that for mCherry (between the NotI and BglII restriction sites; vector denoted pNmCherry). The expression of Orai was checked using fluorescence-detection size exclusion chromatography using the mCherry fluorescence signal (e.g. *Figure 7—figure supplement 1*) (*Kawate and Gouaux, 2006*). The cDNA for full-length *Drosophila melanogaster* STIM was cloned into the pNGFP-EU vector using the NheI and PstI restriction sites and has no expression tag. The constructs of Orai are as follows: (1) Orai$_{cryst-RR}$, the construct of Orai used to determine the quiescent structure (*Hou et al., 2012*); amino acids 133–341 with the following mutations: C224S, C283T, P276R and P277R. (2) Amino acids 120–351 of wild type Orai (denoted 'Orai' in *Figure 2* and *Figure 7—figure supplement 1*). (3) Amino acids 120–351 of Orai with the following mutations: R155S, K159S, and K163S (denoted 'SSS Orai' in *Figure 7*).

HEK293 cells were maintained in Dulbecco's Modified Eagle's medium (DMEM, the Media Preparation Core of MSKCC) supplemented with 10% fetal bovine serum (Gibco, catalog A31606-02). Approximately $1.5 \times 10^6$ cells were co-transfected (in a six-well dish) using the Lipofectamine 3000 transfection reagent (Invitrogen, catalog L3000-015) with 2 μg GCaMP6s plasmid (*Chen et al., 2013*) and 0.15 μg Orai-mCherry plasmid and/or 0.6 μg STIM plasmid (using the P3000 reagent according to the manufacturer's protocol). Empty pNmCherry vector was co-transfected with GCaMP6s as a control. Approximately 16 hr after transfection, the cells were trypsinized (Corning, catalog 25–053 Cl), resuspended in FluoroBrite DMEM (GIBCO, catalog A18967-01) supplemented with 2 mM L-glutamine (GIBCO, catalog 25030–081), and seeded to a 384-well plate (~$3 \times 10^4$ cells per well). Approximately 8 hr after seeding, the cells were gently rinsed once with 0Ca solution (10 mM HEPES pH 7.4, 150 mM NaCl, 4.5 mM KCl, 3 mM MgCl2, 0.5 mM EGTA and 10 mM D-glucose). After removing the rinse solution, 20 μl of 0Ca solution was added to each well and the cells were incubated for 30–40 min before fluorescence measurements. Fluorescence measurements were taken using Hamamatsu FDSS at Ex/Em = 480/527 nm. After recording for 5 min, 10 μl 0Ca solution containing thapsigargin (TG) was added to each well to yield a final concentration of 1 μM TG. After a 10 min incubation with TG, 10 μl solution containing 10 mM HEPES pH 7.4, 146 mM NaCl, 4.5 mM KCl, 10 mM CaCl2 and 10 mM D-glucose was added to each well to yield a final $Ca^{2+}$ concentration of~2 mM. Another 8 min of recording was taken after the $Ca^{2+}$ addition. Fluorescence was recorded every 2 s. The Gcamp6s intensity traces were generated from an average ±SEM of fluorescence

reading from three wells. Data for each time point were plotted as $\Delta F/F_0$ in the Y-axis, where F is the measured fluorescence, $F_0$ is the initial fluorescence value, and $\Delta F = F\ F0$.

## Flow cytometry

Murine monoclonal antibodies 17E5 and 13C8 were raised against purified *Drosophila* Orai by the Antibody and Bioresource core facility at Memorial Sloan Kettering Cancer Center. To assess the binding epitopes of these antibodies, constructs of Orai containing mutations or deletions were made within different parts of the channel and antibody binding was assessed by FSEC. These results indicated that 17E5 binds to the M1-M2 loop on the extracellular side of Orai and that 13C8 binds to a C-terminal region on the intracellular side of the channel.

HEK293 cells that had been prepared for GCaMPs $Ca^{2+}$ influx experiments were also used for flow cytometry. Transfected cells were pelleted and resuspended in FluoroBrite DMEM. Primary monoclonal antibodies (17E5 and 13C8, 1:1000 dilution of cell culture supernatants) were added and incubated with the cells for 20 min at room temperature. The cell samples were then washed two times by pelleting and resuspending in FluoroBrite DMEM. An Alexa Fluor 488 conjugated secondary antibody (Invitrogen, catalog A-21202) was incubated with the cells for 10 min at room temperature. Cells were washed two times, resuspended in FluoroBrite DMEM and filtered with a cell strainer (Falcon, catalog 352235) prior to flow cytometry analysis. Flow cytometry measurements were performed with a Fortessa three flow cytometer (BD Biosciences) at the Flow Cytometry Core Facility of MSKCC. Before measurements, unstained and single stained controls were run for spectral compensations. Six-cell samples were run, as indicated in *Figure 7—figure supplement 1*. 100,000 events were recorded for each of the samples. Data analysis was performed using FlowJo software (FlowJo, LLC). First, living cells were gated and selected based on a forward scatter/side scatter (FSC-A/SSC-A) plot. This excluded approximately 20% of the cells. Second, single cells (approximately 75% of the living cells) were selected based on side scatter height and width (SSC-H/SSC-W). And lastly, mCherry-positive cells were selected according to a plot of compensated Alexa Fluor 488/mCherry fluorescence (Alexa Fluor 488/mCherry, *Figure 7—figure supplement 1c*) and analyzed by histogram plot (*Figure 7—figure supplement 1d*).

FSEC was also used to evaluate the expression of Orai constructs that were used for $Ca^{2+}$ influx and flow cytometry experiments. Transfected cells were solubilized in buffer containing 40 mM Tris-HCl pH 8.5, 150 mM NaCl, 10 mM DDM and 1:500 dilution of Protease Inhibitor Cocktail Set III, EDTA free (CalBiochem) at 4°C for 1 hr and then clarified by centrifugation at 20,800 $g$ for 1 hr at 4°C before FSEC analysis. Supernatants were loaded onto a Superose 6 Increase column (10/300 GL; GE healthcare, equilibrated in 10 mM Tris-HCl pH8.5, 150 mM NaCl and 1 mM DDM) and the fluorescence of mCherry was used for detection (*Figure 7—figure supplement 1b*).

## Acknowledgements

We thank Richard Hite, Christopher Lima, Nikola Pavletich, and members of the Long laboratory for discussions and comments on the manuscript. We thank Petrina Georgala, Frances Weiss-Garcia, the other staff of the Flow Cytometry and Antibody and Bioresource core facilities at Memorial Sloan Kettering Cancer Center, and Deguang Liang for assistance with flow cytometry software. This work was supported by NIH Grant R01 GM094273 (to SBL) and a core facilities support grant to Memorial Sloan Kettering Cancer Center (P30 CA008748). Synchrotron radiation facilities were supported by NIH Grants ACB-12002, AGM-12006, P41 GM103403, and S10 RR029205, under Department of Energy Contract DE-AC02-06CH11357. Atomic coordinates, structure factors, and crystallographic phases have been deposited in the Protein Data Bank with accession numbers 6BBF (H206A Orai$_{cryst}$), 6BBG (WT Orai$_{cryst}$), 6BBH (K163W Orai$_{cryst}$, I4$_1$ form), and 6BBI (K163W Orai$_{cryst}$, P4$_2$2$_1$2 form).

## Additional information

### Funding

| Funder | Grant reference number | Author |
| --- | --- | --- |
| National Institutes of Health | R01 GM094273 | Stephen Barstow Long |

| National Institutes of Health | P30 CA008748 | Stephen Barstow Long |

The funders had no role in study design, data collection and interpretation, or the decision to submit the work for publication.

## Author contributions

Xiaowei Hou, Data curation, Formal analysis, Validation, Investigation, Methodology, Writing—original draft, Writing—review and editing; Shana R Burstein, Data curation, Formal analysis, Investigation, Methodology; Stephen Barstow Long, Conceptualization, Data curation, Supervision, Validation, Writing—original draft, Project administration, Writing—review and editing

## Author ORCIDs

Stephen Barstow Long http://orcid.org/0000-0002-8144-1398

## Decision letter and Author response

Decision letter https://doi.org/10.7554/eLife.36758.034
Author response https://doi.org/10.7554/eLife.36758.035

# Additional files

## Supplementary files

• Transparent reporting form
DOI: https://doi.org/10.7554/eLife.36758.024

## Data availability

Atomic coordinates, structure factors, and crystallographic phases have been deposited in the Protein Data Bank with accession numbers 6BBF (H206A Oraicryst), 6BBG (WT Oraicryst), 6BBH (K163W Oraicryst, I41 form), and 6BBI (K163W Oraicryst, P42212 form).

The following datasets were generated:

| Author(s) | Year | Dataset title | Dataset URL | Database, license, and accessibility information |
|---|---|---|---|---|
| Hou X, Burstein S, Long SB | 2018 | H206A Oraicryst | http://rcsb.org/pdb/search/structidSearch.do?structureId=6BBF | Publicly available at the RCSB Protein Data Bank (accession no: 6BBF) |
| Hou X, Burstein S, Long SB | 2018 | WT Oraicryst | http://rcsb.org/pdb/search/structidSearch.do?structureId=6BBG | Publicly available at the RCSB Protein Data Bank (accession no: 6BBG) |
| Hou X, Burstein S, Long SB | 2018 | K163W Oraicryst, I41 form | http://rcsb.org/pdb/search/structidSearch.do?structureId=6BBH | Publicly available at the RCSB Protein Data Bank (accession no: 6BBH) |
| Hou X, Burstein S, Long SB | 2018 | K163W Oraicryst, P42212 form | http://rcsb.org/pdb/search/structidSearch.do?structureId=6BBI | Publicly available at the RCSB Protein Data Bank (accession no: 6BBI) |

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
