## [Decision Letter]

Thank you for submitting your article "Structures reveal opening of the store-operated calcium channel Orai" for consideration by *eLife*. Your article has been reviewed by three peer reviewers, including Richard S Lewis as the Reviewing Editor and Reviewer #1, and the evaluation has been overseen by Richard Aldrich as the Senior Editor. The following individuals involved in review of your submission have agreed to reveal their identity: Christopher Miller (Reviewer #2); Alexander Sobolevsky (Reviewer #3).

The reviewers have discussed the reviews with one another and the Reviewing Editor has drafted this decision to help you prepare a revised submission.

Summary:

Orai channels mediate store-operated calcium entry in nearly all cells and are essential regulators of the immune response, muscle development and function, and cellular activities such as gene expression, motility, and secretion. This paper by Hou et al. builds upon their earlier crystal structure of Orai in the closed state by describing an open state and a potential intermediate state between open and closed. Orai channels bearing the H206A mutation are constitutively open, and their crystal structure indicates an unexpectedly large dilation of the inner pore structure, with the C terminal (M4-ext) helices extending away from the channel rather than engaging in coiled-coil pairs as in the closed structure. They show that this extension is not sufficient to open the channel but may constitute a "latch" that must be released to allow pore dilation and opening. This paper is highly significant for providing insights into the mechanism of pore opening and the first view of the open pore structure of the Orai channel which suggests mechanisms underlying ion permeation and selectivity.

Essential revisions:

All three reviewers were highly enthusiastic about the work, calling it a crystallographic tour-de-force in which the authors make a 7 Å resolution structure speak clearly, and noted the clever usage of NCS averaging and employment of heavy atoms to strengthen the models and identify functionally important regions. No additional major experiments were thought necessary to support the conclusions, but a number of issues were identified that should be addressed in a revised version of the manuscript.

1) Is the H206A Orai_cryst_ channel fully representative of the open state of native Orai activated by STIM binding? The main evidence for this comes from electrophysiological characterization of H134A Orai1 (the human homolog), which is rather sparse: it has a normal inwardly rectifying I/V and reversal potential in 10 mM Ca^2+^, compared to WT (Frischauf et al., 2017). This is a fairly limited characterization, and does not show that the pore is identical to STIM-activated WT Orai. In one reviewer's experience and in published studies (McNally et al., 2012), high Ca^2+^ levels mask moderate changes to the selectivity filter, which are easily detected under divalent-free conditions from measurements of the Na/Cs permeability ratio and K1/2 for Ca^2+^ block. Given the lack of detailed information about permeation in H206A channels, the authors should acknowledge a degree of uncertainty about how closely the H206A resembles STIM-activated WT Orai and note that a more complete characterization will have to be done to firmly assign this structure to the naturally-occurring open state.

2) Is the WT Orai_cryst_ structure, with its straightened M4-ext helices and closed pore, truly an intermediate in the activation sequence? This argument rests heavily on the idea that the "quiescent" form of Orai from the previous paper (with crossed M4-ext) is a naturally occurring state that somehow has to straighten the M4-ext's before it can bind STIM and open. However, the previous study made mutations in the 3-4 loop (P276R/P277R) that change the conformation of the M4s (Figure 8D) and help crystal formation through interactions of the crossed M4-ext's downstream of the coiled-coil. This raises the question of whether the crossed conformation occurs naturally or is a consequence of the mutations. In this regard, it would be good to know if the previous mutant could be activated by STIM, implying that it functions normally. Some discussion of this point, and the influence of the mutations in promoting the crossed configuration, would be helpful. An alternative view would be that the straight M4-ext conformation represents the true resting state of the channel and not an intermediate.

3) The repositioning of Ba^2+^ in the open structure is significant. The authors should consider discussing this in light of work on the upper gate (Yamashita et al., 2017) which proposed a rotation of the upper part of the M1 helix. Rotation of the side chains could create a favorable site for binding in the plane of the glutamates.

4) Comparison of the Orai open and closed states makes an impression that the hydrophobic region of the pore changes very little with channel opening (Figure 4A). In the closed state, this region apparently blocks permeation as a hydrophobic seal. Does this region open enough for ions to go through? Of course, MD simulations might help to answer this question but given that this study is already overwhelmed with experiments and that MD results might also be ambiguous, can the authors provide additional analysis of their open state structure that might answer this question? For example, measuring the HOLE profile for the open state, similar to the one presented in Figure 1B for the closed state, even under the assumption of the same side chain conformations, might be helpful. Such a profile, for example, will give an opportunity to discuss whether the predicted expansion-type pore opening is wide enough (e.g. compared to pore openings in other ion channels) to support ion permeation or whether it would require rotations of pore-lining helices to "hide" the hydrophobic side chains, as proposed by the experiments of Yamashita et al. (2017).

5) Subsection “X-ray structure of H206A Orai_cryst_ reveals an open conformation”, last paragraph: the authors postulate that the free-energy difference between closed and open may be only a few H bonds. This may apply to the H206A mutant, but is there any indication that the H206-S165 H bond would be broken during normal activation by STIM? Or any mutagenesis on Ser165 that bears upon H-bonding in the closed-open balance?

6) The SSS Orai protein does not conduct constitutively in liposomes, but is this because it is stuck in the closed state or because it fails to assemble properly? The V174A mutant is not a good control here, as it does not have the SSS mutations. It would be fairly straightforward to express SSS Orai with STIM in S2 cells and test for its ability to allow Ca^2+^ influx after ER Ca^2+^ depletion. If it does, one would be able to say that the triple ring of basic residues is not solely responsible for channel closure in the resting state.

[Editors' note: further revisions were requested prior to acceptance, as described below.]

Thank you for resubmitting your work entitled "Structures reveal opening of the store-operated calcium channel Orai" for further consideration at *eLife*. Your revised article has been favorably evaluated by Richard Aldrich (Senior Editor) and a Reviewing Editor.

The manuscript has been improved but there are some remaining issues that need to be addressed before acceptance, as outlined below (numbered as in the original review):

1) The new text here is a bit vague, and I suspect most readers will conclude that the H134A open state is identical to the STIM-induced open state. They are probably not identical, as Yeung et al. (2018) show that STIM1 shifts the Vrev of H134C to more positive potentials (i.e. it is not quite as Ca^2+^-selective as WT). It would be more accurate to describe it as "highly similar" rather than "analogous" to more clearly indicate it is not identical.

2) The new data in Figure 2 showing normal function of Orai_cryst-RR_ in HEK cells are very clean, and the fact that it can be crystallized in the I4_1_ form with a similar cell dimension clearly shows that the RR mutations do not prevent the extension of the M4 helices (subsection “Cloning, expression, purification and crystallization”, last paragraph). These data should be shown as a figure supplement. But the main question here was whether the latched state is really the physiological "quiescent" state, or is induced by the pro to arg mutations. Assuming that WT Oraicryst was never observed in the latched state, please explain in the text why you think the mutations are unlikely to artificially induce the latched state.

4) The new diagram of the pore profile (Figure 11) looks good. But there is no evidence that the widening of the hydrophobic region is "enough to permit ion permeation" by itself. I realize there are uncertainties given the low resolution, so it may be better to just conclude that widening of the hydrophobic region "contributes" to gating, or to relief of the hydrophobic block to permeation. The helical rotation model of Yamashita et al. could be discussed briefly here, to give a fuller picture of other processes that may contribute to gating.

5) The statement about H bonds was not removed from the manuscript, but was moved to Figure 12 legend. If the statement is retained, the logic behind it needs to be explained.

Discussion, first paragraph: The added reference to Stathopulos et al. is not very informative as it does not describe the structure. The most relevant point is that the Stathopulos study proposed STIM binding to a crossed configuration of M4 helices rather than the more extended form described here. Something brief to that effect should be added.

Discussion, third paragraph. With regard to the charge on the E178s, I understand that the charge calculations are imprecise, but a reference or argument should be presented here to explain why you conclude that the pKa values will be "undoubtedly" elevated "dramatically."

There are a number of spelling and grammar errors in the revised text (e.g., text should read "comprises" or "comprising", not "comprised of"). Please check.

---

## [Author Response]

Essential revisions:1) Is the H206A Orai_cryst_ channel fully representative of the open state of native Orai activated by STIM binding? The main evidence for this comes from electrophysiological characterization of H134A Orai1 (the human homolog), which is rather sparse: it has a normal inwardly rectifying I/V and reversal potential in 10 mM Ca^2+^, compared to WT (Frischauf et al., 2017). This is a fairly limited characterization, and does not show that the pore is identical to STIM-activated WT Orai. In one reviewer's experience and in published studies (McNally et al., 2012), high Ca^2+^ levels mask moderate changes to the selectivity filter, which are easily detected under divalent-free conditions from measurements of the Na/Cs permeability ratio and K1/2 for Ca^2+^ block. Given the lack of detailed information about permeation in H206A channels, the authors should acknowledge a degree of uncertainty about how closely the H206A resembles STIM-activated WT Orai and note that a more complete characterization will have to be done to firmly assign this structure to the naturally-occurring open state.

Thank you. We agree and now acknowledge this in the revised manuscript (in the Introduction and the “X-ray structure of H206A Orai_cryst_ reveals an open conformation” section).

2) Is the WT Orai_cryst_ structure, with its straightened M4-ext helices and closed pore, truly an intermediate in the activation sequence? This argument rests heavily on the idea that the "quiescent" form of Orai from the previous paper (with crossed M4-ext) is a naturally occurring state that somehow has to straighten the M4-ext's before it can bind STIM and open. However, the previous study made mutations in the 3-4 loop (P276R/P277R) that change the conformation of the M4s (Figure 8D) and help crystal formation through interactions of the crossed M4-ext's downstream of the coiled-coil. This raises the question of whether the crossed conformation occurs naturally or is a consequence of the mutations. In this regard, it would be good to know if the previous mutant could be activated by STIM, implying that it functions normally. Some discussion of this point, and the influence of the mutations in promoting the crossed configuration, would be helpful. An alternative view would be that the straight M4-ext conformation represents the true resting state of the channel and not an intermediate.

Thank you for these comments. In the revised manuscript, we refer to the conformation with straightened M4/M4-ext helices and a closed pore as the “unlatched-closed” conformation. We now include data showing that the previously-crystallized construct from Hou et al. (2012), which contains the P276R/P277R mutations, functions as a CRAC channel that can be activated by STIM (Figure 2). We also note that the P276R/P277R construct also crystallizes in the same crystal form as the “unlatched-closed” conformation, and this indicates that Orai with the P276R/P277R mutations can have straightened M4/M4-ext helices. The structures indicate that the M4/M4-ext helices can adopt multiple conformations due to different degrees of bending at the Pro288 and SHK hinges. We hypothesize that the conformations of the M4/M4-ext helices are quite dynamic in the cell and that the conformations observed in the crystal structures represent only snapshots of particular conformations within an ensemble of possible ones. Nevertheless, our data indicate that unlatching (which we define as the unpairing of adjacent M4-ext helices and the release of the M4b-M3 interaction, and not necessarily complete straightening of M4/M4-ext) is necessary to allow the opening of the pore that is observed in the H206A open structure.

3) The repositioning of Ba^2+^ in the open structure is significant. The authors should consider discussing this in light of work on the upper gate (Yamashita et al., 2017) which proposed a rotation of the upper part of the M1 helix. Rotation of the side chains could create a favorable site for binding in the plane of the glutamates.

Whether the M1 helix rotates slightly when the channel opens is interesting. However, it is not possible to discern this from our data. Subtle widening or shifting of the glutamates, which may or may not involve helix rotation, could cause the repositioning of Ba^2+^.

4) Comparison of the Orai open and closed states makes an impression that the hydrophobic region of the pore changes very little with channel opening (Figure 4A). In the closed state, this region apparently blocks permeation as a hydrophobic seal. Does this region open enough for ions to go through? Of course, MD simulations might help to answer this question but given that this study is already overwhelmed with experiments and that MD results might also be ambiguous, can the authors provide additional analysis of their open state structure that might answer this question? For example, measuring the HOLE profile for the open state, similar to the one presented in Figure 1B for the closed state, even under the assumption of the same side chain conformations, might be helpful. Such a profile, for example, will give an opportunity to discuss whether the predicted expansion-type pore opening is wide enough (e.g. compared to pore openings in other ion channels) to support ion permeation or whether it would require rotations of pore-lining helices to "hide" the hydrophobic side chains, as proposed by the experiments of Yamashita et al. (2017).

We now include a figure showing the approximate dimensions of the open pore based on the atomic coordinates (Figure 11). Although there is a large degree of uncertainty in this analysis due to the low resolution of the diffraction data, the widening of the hydrophobic region appears to be sufficient to allow ion flow. Due to the limited resolution of the data, we prefer to refrain from speculating further. The relatively narrow hydrophobic region of Orai in comparison to other open channel pores may be one reason that Orai conducts ions considerably slower than most other ion channels.

5) Subsection “X-ray structure of H206A Orai_cryst_ reveals an open conformation”, last paragraph: the authors postulate that the free-energy difference between closed and open may be only a few H bonds. This may apply to the H206A mutant, but is there any indication that the H206-S165 H bond would be broken during normal activation by STIM? Or any mutagenesis on Ser165 that bears upon H-bonding in the closed-open balance?

We do not intend to speculate on possible changes in hydrogen bonding and have removed the phrase from the revised manuscript.

6) The SSS Orai protein does not conduct constitutively in liposomes, but is this because it is stuck in the closed state or because it fails to assemble properly? The V174A mutant is not a good control here, as it does not have the SSS mutations. It would be fairly straightforward to express SSS Orai with STIM in S2 cells and test for its ability to allow Ca^2+^ influx after ER Ca^2+^ depletion. If it does, one would be able to say that the triple ring of basic residues is not solely responsible for channel closure in the resting state.

We have included a substantial amount of additional data on SSS Orai (Figure 7, and Figure 7—figure supplement 1). Purified SSS Orai appears to be properly folded as judged by SEC (Figure 7—figure supplement 1A). The V174A mutant is only included to demonstrate that the assay (Figure 7A) is capable of detecting Na^+^ flux. We now show that SSS Orai does not function as a CRAC channel when co-expressed with STIM in HEK239 cells (Figure 7B). This somewhat surprising result is discussed briefly.

[Editors' note: further revisions were requested prior to acceptance, as described below.]

The manuscript has been improved but there are some remaining issues that need to be addressed before acceptance, as outlined below (numbered as in the original review):1) The new text here is a bit vague, and I suspect most readers will conclude that the H134A open state is identical to the STIM-induced open state. They are probably not identical, as Yeung et al. (2018) show that STIM1 shifts the Vrev of H134C to more positive potentials (i.e. it is not quite as Ca^2+^-selective as WT). It would be more accurate to describe it as "highly similar" rather than "analogous" to more clearly indicate it is not identical.

As suggested, the wording in has been changed to “highly similar”.

2) The new data in Figure 2 showing normal function of Oraicryst-RR in HEK cells are very clean, and the fact that it can be crystallized in the I4_1_ form with a similar cell dimension clearly shows that the RR mutations do not prevent the extension of the M4 helices (subsection “Cloning, expression, purification and crystallization”, last paragraph). These data should be shown as a figure supplement. But the main question here was whether the latched state is really the physiological "quiescent" state, or is induced by the pro to arg mutations. Assuming that WT Oraicryst was never observed in the latched state, please explain in the text why you think the mutations are unlikely to artificially induce the latched state.

In the section entitled “Additional X-ray structures reveal an unlatched-closed conformation” we now explain why we think that the pro to arg mutations are unlikely to artificially induce the latched state. Because Figure 2 supports the conclusion that the Orai_cryst-RR_ construct functions normally, and because inclusion of the experiments shown in Figure 2 was requested during the review process, we would like to retain Figure 2 as a main figure.

4) The new diagram of the pore profile (Figure 11) looks good. But there is no evidence that the widening of the hydrophobic region is "enough to permit ion permeation" by itself. I realize there are uncertainties given the low resolution, so it may be better to just conclude that widening of the hydrophobic region "contributes" to gating, or to relief of the hydrophobic block to permeation. The helical rotation model of Yamashita et al. could be discussed briefly here, to give a fuller picture of other processes that may contribute to gating.

We have removed the phrase “enough to permit ion permeation”, expanded this portion of the Discussion, and incorporated the suggested wording changes.

5) The statement about H bonds was not removed from the manuscript, but was moved to Figure 12 legend. If the statement is retained, the logic behind it needs to be explained.

We had intended to remove the idea and thank the reviewer for careful reading. The sentence has been removed from the figure legend. The idea that we were alluding to is not a radical one – simply that a seemingly innocuous mutation of a histidine to an alanine has a dramatic effect on the conformation of the channel.

Discussion, first paragraph: The added reference to Stathopulos et al. is not very informative as it does not describe the structure. The most relevant point is that the Stathopulos study proposed STIM binding to a crossed configuration of M4 helices rather than the more extended form described here. Something brief to that effect should be added.

We have added a brief discussion of the NMR structure from Stathopulos et al.to the Introduction.Our study says little about how STIM might interact with Orai. The proposed ideas along these lines in the discussion are meant as fodder for thought and are far from a definitive mechanism.

Discussion, third paragraph. With regard to the charge on the E178s, I understand that the charge calculations are imprecise, but a reference or argument should be presented here to explain why you conclude that the pKa values will be "undoubtedly" elevated "dramatically."

Thank you, we now indicate our reasoning along these lines in the Discussion.

There are a number of spelling and grammar errors in the revised text (e.g., text should read "comprises" or "comprising", not "comprised of"). Please check.

Thank you. We have attempted to fix the spelling and grammatical errors.